# Association between resting-state functional brain connectivity and gene expression is altered in autism spectrum disorder

Stefano Berto [1,9], Alex H. Treacher [2,9], Emre Caglayan [1,9], Danni Luo[2], Jillian R. Haney[3,4,5], Michael J. Gandal [3,4,5,6], Daniel H. Geschwind [3,4,5,6], Albert A. Montillo [2,7,8✉] & Genevieve Konopka [1✉]

Gene expression covaries with brain activity as measured by resting state functional magnetic resonance imaging (MRI). However, it is unclear how genomic differences driven by disease state can affect this relationship. Here, we integrate from the ABIDE I and II imaging cohorts with datasets of gene expression in brains of neurotypical individuals and individuals with autism spectrum disorder (ASD) with regionally matched brain activity measurements from fMRI datasets. We identify genes linked with brain activity whose association is disrupted in ASD. We identified a subset of genes that showed a differential developmental trajectory in individuals with ASD compared with controls. These genes are enriched in voltage-gated ion channels and inhibitory neurons, pointing to excitation-inhibition imbalance in ASD. We further assessed differences at the regional level showing that the primary visual cortex is the most affected region in ASD. Our results link disrupted brain expression patterns of individuals with ASD to brain activity and show developmental, cell type, and regional enrichment of activity linked genes.

[1] Department of Neuroscience, UT Southwestern Medical Center, Dallas, TX 75390, USA. [2] Lyda Hill Department of Bioinformatics, UT Southwestern Medical Center, Dallas, TX 75390, USA. [3] Program in Neurobehavioral Genetics, Department of Psychiatry, Semel Institute, David Geffen School of Medicine, University of California, Los Angeles, Los Angeles, CA 90095, USA. [4] Center for Autism Research and Treatment, Semel Institute, David Geffen School of Medicine, University of California, Los Angeles, Los Angeles, CA 90095, USA. [5] Program in Neurogenetics, Department of Neurology, Center for Autism Research and Treatment, Semel Institute, David Geffen School of Medicine, University of California, Los Angeles, Los Angeles, CA 90095, USA. [6] Department of Human Genetics, David Geffen School of Medicine, University of California, Los Angeles, Los Angeles, CA 90095, USA. [7] Department of Radiology, University of Texas Southwestern Medical Center, Dallas, TX, USA. [8] Advanced Imaging Research Center, University of Texas Southwestern Medical Center, Dallas, TX, USA. [9] These authors contributed equally: Stefano Berto, Alex H. Treacher, Emre Caglayan. ✉email: Albert.Montillo@utsouthwestern.edu; Genevieve.Konopka@utsouthwestern.edu

Brain architecture and activity are governed by gene regulatory mechanisms that can be captured using transcriptomic measures[1–3]. How these mechanisms are impacted in neuropsychiatric disorders such as autism spectrum disorder (ASD) remain incompletely understood. Recent advances in human brain imaging genomics have the translational potential to address the challenge of detecting genes associated with either structural or functional measurements[4–6]. For instance, several studies have highlighted the influence of genetic variants on brain imaging phenotypes, identifying common loci that affect brain morphology, structure, and connectivity[7–11]. However, despite this considerable progress in understanding the genetic influence on human brain phenotypes, the gene regulatory mechanisms supporting such functional measurements remain mostly unknown. Identification of such gene expression patterns that underlie functional measures of human brain activity is particularly compelling as such insights will provide opportunities for future modulation of normal or pathological behaviors.

To date, several studies using resting-state functional MRI (rs-fMRI) measurements across cortical regions have identified gene expression patterns that support functional signals in human brain[12–15]. Such studies were a pioneering first step to determine reliable sets of genes that correlate with functional brain network measurements. These studies also established methodologies that can also be applied to study the association between gene expression and functional measurements in neuropsychiatric disorders. For example, genomic perturbations associated with differences in brain activity in a neuropsychiatric disorder such as ASD can now be examined. Individuals with ASD have alterations in both brain activity[16–18], and gene expression patterns (including at the cell-type level)[19–22]; thus, integrating datasets of brain imaging phenotypes, transcriptional landscapes, and cell-type expression patterns should provide insight into ASD pathophysiology. Moreover, because several ASD-relevant genes are chromatin modifiers or involved in neuronal activity[23–26], we hypothesized that brain gene expression patterns that typically support functional brain activity in healthy individuals might be severely affected in ASD. Therefore, coupling measurements of brain gene expression and activity has the potential to identify genes whose expression underlies functional networks observed in rs-fMRI and how such relationships are altered in ASD.

Here, we apply an approach to understand the gene expression signals that may underlie human brain activity (as assessed by rs-fMRI) relevant to ASD. In contrast with previous studies that used a reference dataset from a small number of "control" brain donors[27–29] or blood[30], we use post-mortem brain gene expression datasets from a greater number of individuals who are characterized as either neurotypical or who were diagnosed with ASD. Because of the rarity of post-mortem tissue available from ASD brain donors, our study is restricted to a subset of cortical regions. Nonetheless, we identify genes with expression patterns in brains from individuals with ASD that are differentially correlated with rs-fMRI activity. We also identify a small number of cortical regions that display the greatest impact of gene expression on brain activity (e.g., primary visual cortex and inferior temporal cortex). Our analyses consider the developmental expression pattern of the genes we identify related to ASD status. We find that many of these genes have altered expression patterns over postnatal development into adulthood suggesting that these particular genes are indeed relevant for brain activity responsiveness. Together, our results provide key insights into both specific genes and cortical regions that are at risk in ASD. The coupling of two diverse measurements (transcriptome and rs-fMRI) facilitates the prioritization of specific ASD mechanisms that might be missed by using only one type of dataset.

## Results

**Integration of resting-state functional MRI and gene expression measures in individuals with ASD and controls.** To identify differentially correlated genes, we determined the spatial similarity between rs-fMRI and gene expression changes in the human brain of subjects with ASD compared to controls across 11 matched cortical regions. We used rs-fMRI data from an imaging database containing individuals with ASD and matched controls (ABIDE I[31] and ABIDE II[32]) and cortical RNA-sequencing (RNA-seq) datasets from persons with ASD and matched controls across development into adulthood[33] (Fig. 1). We computed two extensively validated measures of brain activation to characterize brain function from rs-fMRI. The first brain measure, fractional Amplitude of Low-Frequency Fluctuations (fALFF)[34], quantifies a subset of brain activity within the low frequency band that form a fundamental feature of the resting brain, and that activity is vitally important whether at rest (daydreaming, musing) or attending to a specific task. The second brain measure, Regional Homogeneity (ReHo)[35], is a complementary measure of the similarity in the temporal activation pattern manifested by clusters of voxels rather than single voxels as in fALFF. This measure of local functional connectivity is itself a close derivative of the underlying brain activity[35]. We generated voxel-wise maps of fALFF and ReHo for a total of 1983 subjects from the ABIDE I and ABIDE II datasets (ASD = 916, CTL = 1067; Supplementary Fig. 1 and Supplementary Data 1), and analyzed a total of 11 regions of interest (ROIs) matching the transcriptomic data using Brodmann area (BA) designations: BA1/2/3/5 (somatosensory cortex), BA4/6 (premotor and primary motor cortex), BA7 (superior parietal gyrus), BA9 (dorsolateral prefrontal cortex), BA17 (primary visual cortex), BA20/37 (inferior temporal cortex), BA24 (dorsal anterior cingulate cortex), BA38 (temporal pole), BA39/40 (inferior parietal cortex), BA41/42/22 (superior temporal gyrus), BA44/45 (inferior frontal gyrus).

We first assessed differences between cases and controls for both fALFF and ReHo (Fig. 2a). We identified 4 ROIs with a significant difference for fALFF and 1 ROI for ReHo (Wilcoxon Rank Sum Test, $p < 0.05$; Supplementary Fig. 2a). BA20/37 was commonly different using either measurement. Even though effect sizes were small between cases and controls for all the ROIs analyzed (Cohen's $d$; $d < 0.3$) in agreement with other reports[36,37], we observed consistency between fALFF and ReHo (Spearman Rank Correlation, rho = 0.46; Fig. 2b). These data reflect subtle, yet replicable functional activity measurements linked to ASD calculated by two rs-fMRI measurements. However, because the differences between cases and controls using rs-fMRI were minimal with a small to null contribution to the analysis, we assessed the rs-fMRI—gene expression relationship using the control subject ReHo and fALFF values. We first assessed the complementarity of these two rs-fMRI measurements in controls. There was a significant correlation between fALFF and ReHo values across individuals in each singular ROI analyzed (Spearman's rho = 0.58, $p < 2.2 \times 10^{-16}$; Supplementary Fig. 3a, b). These data further confirmed the complementarity of these two distinct measurements of rs-fMRI values. To understand ASD pathophysiology in the context of brain activity and gene expression, we spatially matched RNA-seq data[33] from 11 cortical areas for a total of 360 tissue samples from cases (ASD) and 302 control samples (CTL) (Supplementary Fig. 4a). The variance explained by technical and biological covariates was accounted for and removed before further analyses (see "Methods" and Supplementary Fig. 4b).

**Identification of genes differentially correlated with rs-fMRI between ASD and controls.** We sought to identify genes with correlated expression to imaging measurements across regional

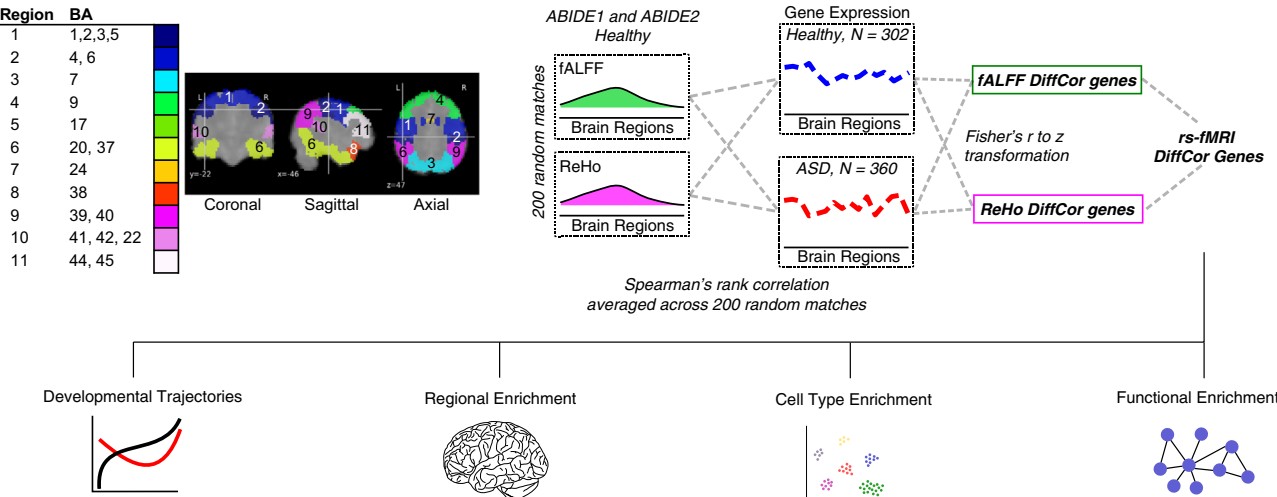

**Fig. 1 Flowchart of the analytical framework and pipeline.** Gene expression values were obtained for 11 cortical regions from individuals diagnosed with autism spectrum disorder (ASD) and demographically matched neurotypical individuals (CTL: control). For fMRI, the ABIDE I and II datasets were used to select ASD and CTL individuals that demographically matched with the gene expression cohort. Regional Homogeneity (ReHo) and fractional amplitude of low frequency oscillations (fALFF) were calculated from these individuals for the 11 cortical regions. 200 subsampled matches were selected from CTL individuals. Spearman's rank correlation was used to infer a correlation between gene expression and fMRI separately for both fMRI measurements. For differential correlation (DC genes) between ASD and CTL, the mean correlation values of matches were transformed to a z score (Fisher's r to z transformation) and statistics combined (Fisher's combined method). A gene was called differentially correlated if the differential correlation p-value was less than 0.01 (DiffCorP < 0.01) and FDR < 0.05 in CTL. The final list consisted of genes that are differentially correlated using both fMRI measurements.

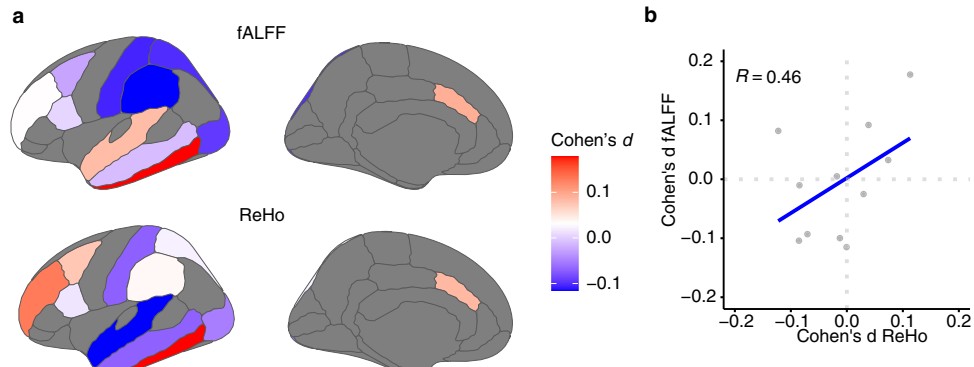

**Fig. 2 Imaging differences between ASD and CTL. a** Differences between ASD and CTL calculated by Cohen's d (effect sizes) derived from ASD–CTL comparison for both rs-fMRI measurements across the ROIs analyzed. **b** Scatter plot depicting the spatial correlation between Cohen's d values of fALFF and ReHo. Each dot corresponds to the ROI analyzed.

rs-fMRI values. To do this, we used Spearman's rank correlation between mean regional values of fALFF or ReHo and regional gene expression. To take advantage of the entire ABIDE dataset, we randomly sampled from the ABIDE dataset 200 times, and correlated each sample with the genomic data (see "Methods"). We defined genes correlated with ReHo and/or fALFF in both controls and ASD (Supplementary Fig. 5a and Supplementary Data 2). Using a Fisher r-to-z transformation, we assessed the significance of the difference between ASD and CTL correlations in both fALFF or ReHo values. We next used a Fisher's method to combine the resultant p-values defining 415 differentially correlated genes (DC genes; Diff Cor P < 0.01, CTL FDR < 0.05; Fig. 3a; "Methods"). DC genes showed a high proportion of positively correlated genes with similar correlation coefficients in both measurements (59.8%; Fig. 3b; Supplementary Fig. 5b). We next examined the effect sizes and the relationship between fALFF and ReHo values (Fig. 3c; Supplementary Fig. 5c). For a P < 0.01, DC genes showed an effect size larger than 1.8, resulting in ~3% of the

gene expressed in our data (Fig. 3c). Among the genes with highest effect size, we found *FILIP1*, which encodes a filamin A binding protein important for cortical neuron migration and dendrite morphology[38–40], and *GABRQ*, a gene encoding a GABA receptor subunit highly expressed in von Economo neurons[41,42]. In addition, the effect sizes of the DC genes calculated with fALFF and ReHo strongly correlate (Spearman Rank Correlation, rho = 0.54, p < 2.2 × 10⁻¹⁶; Supplementary Fig. 5c), further confirming the reproducibility of the DC genes in two different rs-fMRI measurements.

Next, we compared the genes we identified with genes linked with rs-fMRI values from independent studies[14,15]. Because these earlier studies analyzed only healthy individuals, we first compared the genes correlated only in CTL with the ones previously reported. We found that previously fMRI-correlated genes were significantly enriched in CTL genes, revealing reproducibility of fMRI-correlated genes despite variation in cortical regions and type of fMRI measurements (Wang et al.:

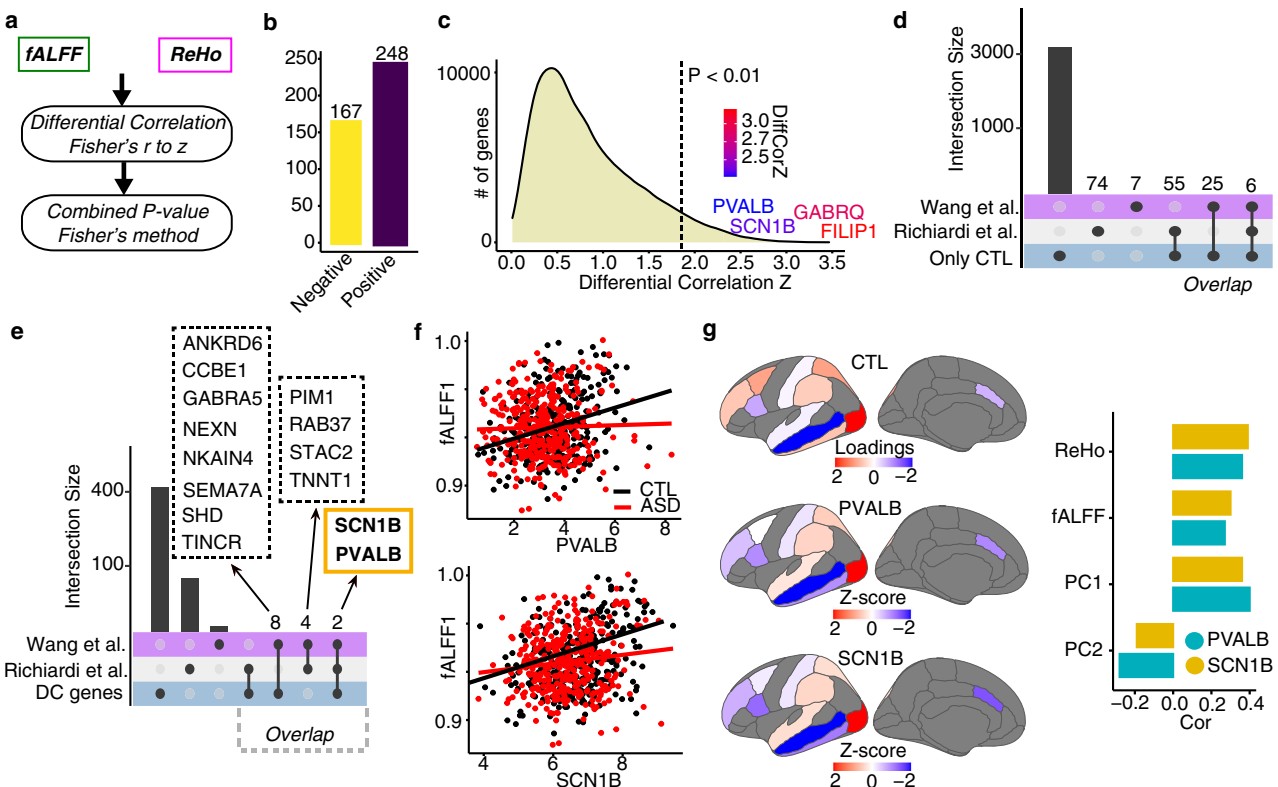

**Fig. 3 Overview of differentially correlated genes. a** Schematic workflow to define differentially correlated genes. **b** Barplot depicting the number of differentially correlated genes positively (purple) and negatively (yellow) correlated with CTL. **c** Density plot depicting the distribution of differential correlation effect sizes (z). Line corresponds to the p value cutoff used. Genes of interest are highlighted with a gradient color that reflects the relative effect size. P-value threshold for Differential Correlation analysis is shown based on Fisher's combined P-value. **d** Upset plot showing the intersection between the genes significantly correlated only in CTL for both fALFF and ReHo and two previous rs-fMRI—gene expression studies using only data from neurotypical individuals. **e** Upset plot showing the intersection between DC genes and two previous rs-fMRI—gene expression studies using only data from neurotypical individuals. **f** Scatterplots showing the relationship (Spearman's rank correlation) between rs-fMRI (Y-axis) and gene expression for two candidate genes (X-axis) in CTL and ASD. **g** Gradient of CTL expression (PC1), PVALB, and SCN1B gene expression. Bar plot depicts the correlation between PVALB and SCN1B gene expression with ReHo, fALFF, expression PC1, and expression PC2.

odds ratio (OR) = 17.3, FDR = $2.5 \times 10^{-15}$, Richiardi et al.: OR = 3.2, FDR = $1.5 \times 10^{-10}$; Fig. 3d, Supplementary Fig. 5d). Among them, 6 genes overlapped in all previous studies, and 2 out of the 6 genes were also among the DC genes between ASD-CTL (*PVALB and SCN1B*; Fig. 3d–f). These two genes are particularly compelling as *SCN1B*, which encodes a beta-1 subunit of voltage-gated sodium channel, is a highly expressed gene in fast-spiking parvalbumin (PVALB+) cortical interneurons, which play a key role in neuronal networks, and whose oscillations are linked with ASD[43–46]. Because PVALB gene expression has a rostrocaudal axis gradient[47], we next evaluated the spatial distribution of both candidates' gene expression in the ROIs. We found that both PC1 (principal component 1), as well as SCN1B and PVALB, displayed differences in the rostrocaudal axis (*PVALB* ~ PC1, *rho* = 0.41; *SCN1B* ~ PC1, *rho* = 0.37), with higher expression in caudal cortical regions (Fig. 3g). These genes were similarly correlated with rs-fMRI measurements, (*PVALB* ~ fALFF, *rho* = 0.32; *SCN1B* ~ fALFF, *rho* = 0.33; *PVALB* ~ ReHo, *rho* = 0.38; *SCN1B* ~ ReHo, *rho* = 0.42), but these correlations are affected by ASD status (Fig. 3f). Overall, these results identify many brain activity-related genes and imply that some of the high confidence genes such as *PVALB* and *SCN1B* support brain activity affected in ASD.

**Differentially correlated genes have specific developmental trajectories.** Although we identified DC genes across all samples

with a median age of 22 years old, we asked how DC genes compare between CTL and ASD across development given that autism is a neurodevelopmental disorder. We leveraged the transcriptomic dataset from this study to detect whether DC genes follow a specific developmental trajectory in individuals with ASD compared with CTL subjects (see "Methods"). We identified three main clusters of DC genes: one highly expressed in adults (Adult), one highly expressed in early development (EarlyDev), and one with relatively stable trajectory throughout development (Stable) (Fig. 4a). Interestingly, genes in the Adult cluster are upregulated until adulthood in neurotypical individuals but this upregulation is delayed in individuals with ASD. In contrast, the genes in the Stable and EarlyDev clusters follow a similar trajectory in both groups (Fig. 4a and Supplementary Fig. 6a). Because each region differs by sample size, we used a subsample approach and recalculated the developmental trajectories. We found that differences in sample size between regions did not affect the overall result (Supplementary Fig. 6b). We additionally assessed gene expression patterns in BrainSpan dataset[48] generated using healthy brain tissue (0–40 years old) and found similar trajectory patterns with the Adult cluster displaying immediate upregulation until adulthood similar to CTL in our dataset (Supplementary Fig. 6c).

Next, we sought to understand the functional properties of the genes associated with these developmental trajectories. Overall, we found enrichments for transporter activity, ion channel activity, and DNA-binding activity which are crucial for proper

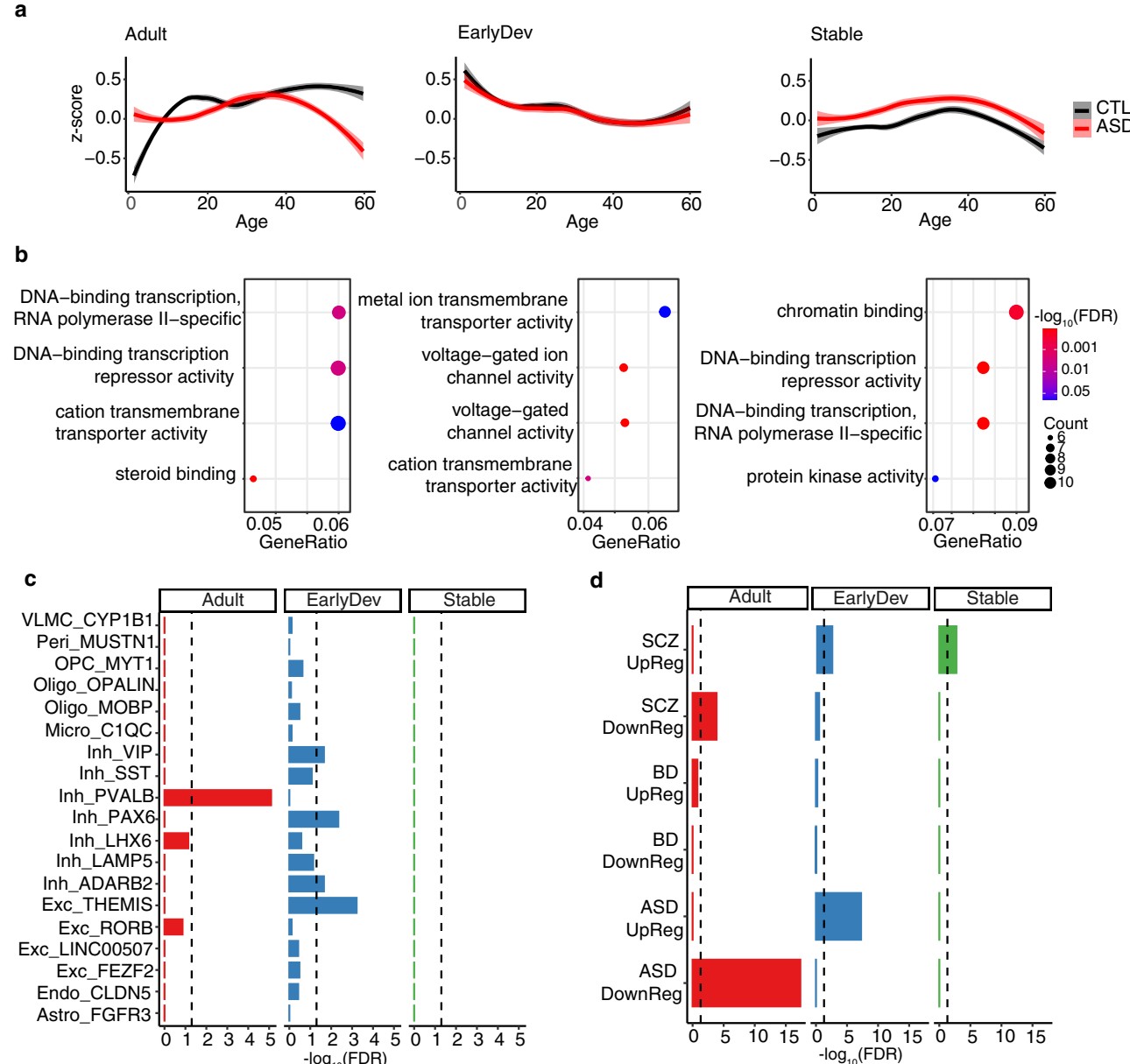

**Fig. 4 Differentially correlated genes in individuals with ASD are important for brain development. a** The 415 genes were clustered in three developmental time groups: adult (Adult), early development (EarlyDev), stable (Stable). X-axis represents developmental time. Y-axis represents the expression based on human brain developmental time. Loess regression with confidence intervals depicts the overall distribution. Smooth curves are shown with 95% confidence bands with relative trendlines. **b** Bubblechart representing the functional enrichment for modules associated with developmental time. Y-axis corresponds to the odds ratio, X-axis corresponds to the $-\log_{10}$(FDR). **c** Bar plot depicting the $-\log_{10}$(FDR) of the Fisher's exact test enrichment between developmental clusters and cell-type markers (Y-axis) from single-nuclei RNA-seq from multiple brain regions ("Methods"). VLMC=vascular leptomeningeal cell, Peri=pericytes, OPC=oligodendrocyte precursor cell, Micro=microglia, Inh=inhibitory neuron, Exc=excitatory neuron, Endo=endothelial cell, Astro=astrocyte. **d** Bar plot depicting the $-\log_{10}$(FDR) of the Fisher's exact test enrichment between developmental clusters and genes differentially regulated in ASD, schizophrenia (SCZ), and bipolar disorder (BD) (Y-axis) from an independent study ("Methods").

development and have been repeatedly implicated in ASD[25,49,50] (Fig. 4b and Supplementary Data 2). However, these enrichments were not distinct for a single developmental trajectory. In contrast, enrichment in steroid binding was only present in the Adult cluster with relatively high significance (Fig. 4b). Steroid binding was mainly driven by enrichment for estrogen receptor (*ESRRG*, *ESRRA*) and nuclear glucocorticoid receptor (*NR3C1*, *NR3C2*) genes. We find this intriguing given that steroid levels are altered in autistic individuals even in early development[51,52] and the ratio of sexes was very similar in our dataset (CTL female ratio: ~0.18, ASD female ratio: ~0.18). Thus, our results indicate

that altered steroid biology in ASD is linked to brain activity changes across cortical regions.

To understand the cell type-specific properties of the rs-fMRI genes, we performed enrichment for gene expression data derived from single-cell RNA-seq studies[41] ("Methods"). We observed that the genes in the Adult cluster were highly enriched for parvalbumin (*PVALB*) expressing interneurons whereas EarlyDev genes were enriched for excitatory neurons. No cell-type enrichment was detected for the genes in the Stable cluster (Fig. 4c). Because PVALB expression follows an anterior to posterior regional gradient[47], we imputed *PVALB*+ interneurons

abundance for each of the region analyzed (see "Methods"). We conducted a deconvolution analysis using MuSiC[53], which allows the inference of relative cell-type abundance in bulk data. Single-nuclei RNA-seq from a multi-cortical region data was used to infer cell-type proportions[54]. We estimated the relative cell-type abundance by subjects and brain regions. As expected, the *PVALB*+ interneurons fractional abundance was higher in posterior regions compared with anterior regions (Supplementary Fig. 6d; Supplementary Data 3). Notably, the relative abundance of these interneurons was significantly reduced in individuals with ASD in posterior regions such as BA7 and BA17. These data indicate that our results are potentially driven by *PVALB*+ interneurons regional abundance further demonstrating the important role of these interneurons in ASD pathology. We next investigated the association of developmental gene clusters with genomic data from brain disorders including ASD[55]. The Adult cluster is enriched for downregulated genes in individuals with ASD while the EarlyDev cluster is enriched for upregulated genes in individuals with ASD (Fig. 4d). This result was relatively specific to ASD as similar gene lists from individuals with schizophrenia or bipolar disorder showed little to no enrichment (Fig. 4d). This result was further confirmed using modules of co-expressed genes dysregulated in such disorders (Supplementary Fig. 6e). Together, these data extend the emerging picture of molecular pathways disrupted in ASD corresponding to rs-fMRI measurements[14,15,30,56].

**The relationship of rs-fMRI and gene expression is altered at the brain region level.** Due to the limited number of samples per ROI, we were not able to assess the association between brain activity and gene expression at a regional level. We overcame this limitation with a leave-one-region out (LoRo) approach inferring the contribution of each region in our results (see "Methods"). Briefly, by leaving one of the 11 regions out at a time, we were able to test whether the differential correlation was affected by one region or several specific regions. We calculated the z-score from the z-to-r Fisher transformation from each analysis and combined with the Fisher's method (Supplementary Data 4). We observed a significant contribution from the primary visual cortex (BA17), temporal cortex (BA20/37, BA38), parietal cortex (BA39/40), and motor cortex (BA4/6) (Fig. 5a). We next examined the enrichment of regional differential expressed genes (DEG; FDR < 0.05, |log$_2$(FC)| > 0.3; see "Methods") between ASD-CTL in DC genes. We explored whether any of the developmental gene clusters were enriched for specific regional DEG. Interestingly, we found the highest enrichment of Adult and EarlyDev cluster genes in cortical areas associated with vision and proprioception (BA17 and BA7) (Fig. 5b). Taken together, these results support the emergent role of the visual cortex in ASD pathophysiology[57,58].

## Discussion

Assessing gene expression in the brain permits a relevant examination of how biological pathways might be altered in the tissue of interest and connected to genetic predispositions. Moreover, functional imaging provides an important window into phenotypes associated with mental illness. Combining these approaches can help begin to bridge the gap between genes and behavior. Indeed, previous work has demonstrated a correspondence between human brain gene expression and functional connectivity as measured by fMRI[14,15,30]. However, the studies using brain gene expression only used neurotypical populations. Local brain activity measures such as ReHo and fALFF can assess neuronal connectivity and activity. When restricted to a specific image acquisition site and age range (e.g., children or adolescents), previous studies using ReHo and fALFF have found

significant differences between CTL and individuals with ASD in cortical regions but in different brain regions and directions[59–63]. However, protocol variability across sites can induce inconsistent findings in functional connectivity[64]. A quantitative meta-analysis indicated that only connectivity between the dorsal posterior cingulate cortex and the right medial paracentral lobule consistently differs between individuals with ASD and CTL subjects across sites and ages;[65] however, these regions were not available for tissue sampling in this study. Structural imaging studies have also indicated the difficulty in finding differences between individuals with ASD and CTL subjects when no age restriction is imposed[66–68]. In contrast to these age and site-restricted reports, our study includes ages from 5 to 64 years and data from 37 sites whose differences are retrospectively normalized and such differences with previous studies likely underlie our finding of few significant differences in brain activity between cases and controls.

We speculated that the expression of genes and their association with brain activity may underscore their potential relevance for any functional brain activity that is disrupted in ASD. In line with this, our results suggest that genes typically associated with rs-fMRI lose their association when ASD genomics are included. These genes are important for brain development, regional differences, and excitatory/inhibitory identity. As previously reported, GABAergic signaling is disrupted across mouse models of ASD[69] and GABA interneurons have a key role for cortical circuitry and plasticity[70–72]. Interestingly, genes highly expressed in the Adult gene cluster that are significantly associated with brain activity are overrepresented in a subpopulation of inhibitory interneurons expressing parvalbumin. In contrast, genes highly expressed in early development are overrepresented in excitatory neurons. In line with the role of parvalbumin neurons in normal brain circuitry and oscillations[70,73,74], this distinct association might underscore the excitation-to-inhibition ratio imbalance in autism. Moreover, the relative abundance explained by the Adult genes of *PVALB*+ interneurons is significantly decreased in individuals with ASD. Because spatial *PVALB*+ expression covaries with rs-fMRI across regions[47], we hypothesized that the relative increased abundance of these interneurons in the visual cortices and conversely the reduction shown in individuals with ASD explains the differential correlation in the specific subset of Adult genes. Therefore, these results further underscore the important role of parvalbumin interneurons in autism.

We hypothesize that genes severely dysregulated in autism such as *SCN1B, KCNAB3, FMN1, or VAMP1* might additionally contribute to the excitation-to-inhibition ratio affecting normal network function and circuitry. Moreover, previous studies have shown that inhibitory neurons control visual response precision with increased activity leading to a sharpening of feature selectivity in mouse primary visual cortex[58]. Additionally, multiple lines of evidence have indicated that individuals with ASD show slower switching between images in binocular rivalry[57,75–77]. Here, we provide evidence that regional brain expression influences the association between rs-fMRI values and gene expression, with the visual cortex as the major contributor to the variance explaining the rs-fMRI—gene association. Therefore, these results contribute to a consistent emerging role of the visual cortex in ASD pathology. However, because subjects who underwent fMRI measurements might not have had uniform instructions (or resultant behavioral compliance) to keep their eyes open or closed, it is possible that the visual cortex data could be influenced by such behavior.

Finally, any functional interpretation of the genes identified should be made with caution. Here, we assessed the relationship between gene expression and rs-fMRI across cortical areas based on correlation, which is not necessarily evidence of causation.

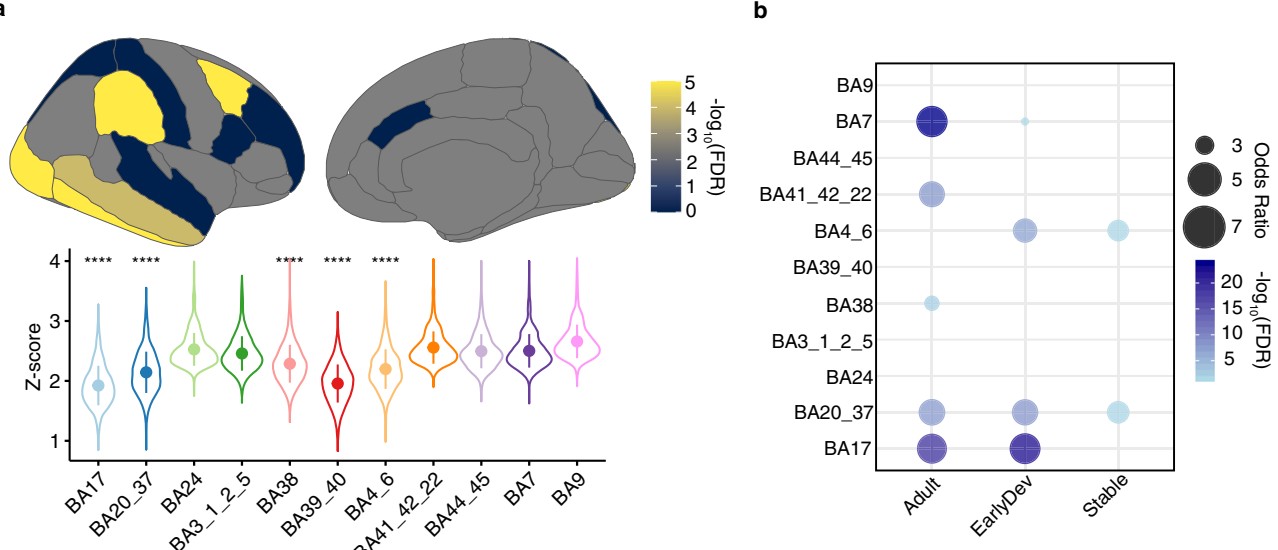

**Fig. 5 Leave one region out (LoRo) analysis underscores the importance of specific brain regions. a** Brain visualizations and violin plots depicting the contribution of each ROI in the differential correlation after the specific region was removed (LoRo). Brain visualizations represent the $-\log_{10}$(FDR) of the comparative analysis between ROIs after LoRo analysis (One-sided Wilcoxon's rank sum test). Violins represent the Z-score ($Y$-axis) of differential correlation after the specific region ($X$-axis) was removed. **** corresponds to a significant lower Z-score compared with other regions ($p < 0.001$; One-sided Wilcoxon's rank sum test). Dots represent the mean Z-score for the specific Brodmann area. Lines represent the standard deviation (SD). $N = 415$ genes from independent analysis. Exact $P$-value: BA17 $p < 2e^{-16}$, BA20_37 $p < 2e^{-16}$, BA38 $p = 6.8e^{-05}$, BA39_40 $p < 2e^{-16}$, BA4_6 $p < 2e^{-16}$. **b** Bubblechart with $-\log_{10}$(FDR) and Odds Ratio from Fisher's Exact test representing enrichment between developmental groups and genes differentially expressed in each region. $X$-axis shows abbreviations for each region. $Y$-axis represents each developmental cluster identified.

Moreover, functional imaging analysis of the ROIs used does not show large differences between neurotypical and individuals with autism. The primary limiting factor in spatial resolution and brain coverage is driven by the restricted tissue sampling/availability from postmortem disease cohorts. Larger sample sizes may in the future allow for a more detailed investigation of these genes at the regional level increasing both specificity and sensitivity. Additionally, candidate genes should be further analyzed in vivo using model systems to provide a basic understanding of their effects on brain activity. In conclusion, we have established that autism pathology significantly impacts the relationship between gene expression and functional brain activity. Our results uncovered genes that are important for excitation-to-inhibition ratio balance and visual cortex function. These results provide molecular mechanisms for future studies relevant to understanding brain activity in individuals with autism.

## Methods

All research in this manuscript complies with all relevant ethical regulations. This study was approved by the UT Southwestern Medical Center Institutional Review Board.

**fALFF and ReHo.** To provide image-derived phenotypes (IDPs) for each subject in the ABIDE cohort, regional measures of brain function were computed including the fractional amplitude of low-frequency fluctuation (fALFF; https://fcp-indi. github.io/docs/latest/user/alff.html?highlight=falff) and regional homogeneity (ReHo; https://fcp-indi.github.io/docs/latest/user/reho). Supplementary Fig. 1 illustrates the main processing steps of the image analysis pipeline.

**Imaging materials.** This study used resting-state fMRI from the 916 ASD and 1067 CTL subjects of both ABIDE I and ABIDE II[32,78]. Details of the pulse sequence parameters used in this data acquisition are provided in Supplementary Data 1. After the removal of subjects with image artifacts, high head movement, or poor MNI152 coregistration, we analyzed the data from the remaining 710 ASD (79% male), and 606 CTL (87% male) subjects, whose age ranges from 5 to 64 years.

**fMRI preprocessing.** All data from each subject were preprocessed consistently—as described below—and are illustrated in Supplementary Fig. 1. The 3dSkullStrip method from the brain extraction tool (BET) was applied to remove skull and non-brain tissue[79]. The first 5 volumes were censored to allow for MRI scanner dynamic instability to settle. To correct for head movement, volume realignment was applied frame by frame, to register each volume to the mean volume with an affine transformation. Slice timing correction was applied to ensure volume slices align temporally.

Images were processed with a generalized linear model (GLM) to regress out: (1) global signal fluctuation, (2) physiological noise represented by white matter and CSF fluctuation, (3) fluctuation correlated with the 6 original affine head motion parameters (X/Y/Z/pitch/roll/yaw), (4) their first derivatives, squares, and squared derivatives, and 5) noise fluctuations captured from five components from aCompCor[80]. Scrubbing was applied to remove frames with a Jenkinson framewise displacement (FWD) > 0.5 mm, and subsequently replaced with an interpolated frame. ReHo was calculated with scrubbed data; however, ALFF and fALFF were not calculated with scrubbed data because the framewise removal and alteration disrupts the temporal structure precluding Fourier transform-based approaches[81].

For subjects with multiple fMRI scans, the scan with the lowest head motion, measured by mean FWD, was selected for analysis. For each resulting subject scan, a subject was excluded if their scan had excessive head motion. Specifically, scans meeting at least one of these three requirements were removed: (1) mean FWD > 0.30 mm, (2) greater than 50% of frames being scrubbed, or (3) scans with outlier mean, 1st, 2nd, or 3rd quantile DVARS values. DVARS was defined as the root mean square of the temporal change of the fMRI voxel-wise signal at each time point[82,83]. The package CPAC v1.8.0 was used for fMRI pre-processing including head motion correction, scrubbing, and nuisance regression.

**Calculation of fALFF and ReHo.** We computed fALFF and ReHo from the resting-state fMRI using C-PAC (v1.8.0)[84] in native subject space, resulting in a volumetric map of fALFF and a map of ReHo for each subject. fALFF[34] quantifies the slow oscillations in brain activity that form a fundamental feature of the resting brain. ALFF is defined as the total power within the low-frequency range (0.01–0.1 Hz) and forms an index of the intensity of the low-frequency oscillations. The normalized ALFF known as fALFF is defined as the power within that low-frequency range normalized by the total power in the total detectable frequency range. fALFF characterizes the contribution of specific low-frequency oscillations to the entire frequency range[34]. To increase the signal to noise ratio by removing high-frequency information, we spatially smoothed each derivative map with a Gaussian kernel. ReHo[35] aims to detect complementary brain activity manifest by clusters of voxels rather than single voxels as in fALFF. ReHo evaluates the similarity of the activity time courses of a given voxel to those of neighboring voxels using Kendall's coefficient of concordance (KCC)[85] as the index of time series similarity. This

measure requires the cluster size as an input to define the size of the neighborhood. In this study, we used a cluster size of 27 voxels. The 26 neighbors of a voxel, $x$, are those within a 3 x 3 x 3 voxel cube centered on voxel $x$. The similarity of the activation time courses between each voxel, $x$, and its 26 nearest neighbors was calculated using: $W_x = (\sum(R_i)^2 - n(\bar{R})^2)/(\frac{1}{12}K^2(n^3 - n))$. $W_x$ is the KCC for voxel $x$ and ranges from 0 to 1, representing no concordance to complete concordance. $R_i$ is the rank sum of the $i$th time point. $R$ is the mean value over the $R_i$'s. K is the cluster size for the voxel time series (here $K = 27$). n is the total number of ranks.

**Registration.** The mean processed fMRI image was nonlinearly registered directly to an EPI template in MNI152 space using the symmetric normalization (SyN) non-linear registration method of the ANTs (v2.3.5) package[86,87]. The resulting composite transform was then applied to both the fALFF and ReHo maps to provide derivative maps in normalized MNI152 space. We used EPInorm-based registration as it better accounts for nonlinear B0 field inhomogeneities at the air to tissue interfaces[88,89]. Supplementary Figure 7 illustrates the improvement EPInorm-based registration has over more the commonly applied T1norm based registration. In this study, EPInorm registration yielded more accurate spatial normalization of the brains to the standard atlas space in which regional values are computed. Regions of improved registration included the sinuses which present air/tissue interfaces that induce non-linear distortions which are properly handled through EPInorm co-registration. EPInorm registration also had a substantially lower standard deviation around the brain periphery across the 1316 subjects assessed.

Lastly, subjects with poor EPInorm registration[88] (discussed below) were removed. Specifically, mis-registration was identified through a combination of manual inspection and through the detection of scans with an outlier number of misaligned brain-masked voxels using the interquartile range (IQR) outlier test[90].

**Segmentation.** In this study, we adapted the Brodmann atlas publicly available through MRICron (v1.0.9) to form the 11 multi-area regions from which tissue samples were drawn from matched donor brains. Supplementary Data 1 illustrates how we combined Brodmann areas to generate 11 regions that correspond with the RNA sequence data. We used the resulting 11 region atlas to assign a region label (parcellate) to each voxel in the fALFF and ReHo maps to enable computation of the mean regional fALFF and ReHo values for all subjects.

**Site correction.** We accessed publicly available ABIDE data across 30 different sites. These sites used MRI devices from different manufacturers (Siemens, Philips, GE) and used different MRI pulse sequences and participant protocols, which can cause differences in the absolute value of the fMRI acquired and can affect fALFF and ReHo values (Supplementary Data 1). As the mean fALFF and ReHo varied between sites, we applied a correction to minimize site differences. To suppress site differences, the difference between the cohorts mean regional value and each site's mean regional value was calculated. This regional difference was then subtracted from each region value for all subjects belonging to the corresponding site.

**Derivative map normalization.** To provide better inter-subject comparisons, we normalized regional fALFF and ReHo values to the weighted mean, weighted by the number of voxels for each region, over all of the regional values for each subject. To reduce the impact of confounders, we regressed out age, site, and sex using a linear model.

**RNA-seq processing and analysis.** Quality control was performed using FastQC (v.0.11.9). Reads were aligned to the human hg38 reference genome using STAR[91] (v.2.5.2b). Picard tool was implemented to refine the quality control metrics (http://broadinstitute.github.io/picard/) and to calculate sequencing statistics. Gencode annotation for hg38 (v.25) was used for reference alignment annotation and downstream quantification. Gene level expression was calculated using RSEM[92]. Dup15q individuals were removed from the initial data[33]. Technical replicates were collapsed by the maximum expression value and maximum RNA integrity value. A total of 302 Control and 360 ASD were used for the final analysis. Supplementary Figure 8 represents the pairwise comparison of demographics from the RNA-seq and rsfMRI datasets. Supplementary Data 1 provides details on all the covariates. Only protein-coding genes were considered. Counts were normalized using counts per million reads (CPM) with the *edgeR* (v3.32.0) package in R[93]. Normalized data were log2 scaled with an offset of 1. Genes were considered expressed with $log_2(CPM + 1) > 0.5$ in at least 80% of the subjects. Normalized data were assessed for effects from known biological covariates (*Sex, Age, Ancestry,* and *PMI*), technical variables related to sample processing (*Batch, BrainBank, RNA Integrity value (RIN)*) and technical variables related to sequencing processing based on PICARD statistics (https://broadinstitute.github.io/picard/).

We used the following sequencing covariates: picard_gcbias.AT_DROPOUT, star.deletion_length, picard_rnaseq.PCT_INTERGENIC_BASES, picard_insert.MEDIAN_INSERT_SIZE, picard_alignment.PCT_CHIMERA Spicard_alignment.PCT_PF_READS_ALIGNED, star.multimapped_percent, picard_rnaseq.MEDIAN_5PRIME_BIAS, star.unmapped_other_percent, picard_rnaseq.PCT_USABLE_BASES, star.uniquely_mapped_percent.

Residualization was applied using a linear model. All covariates except Diagnosis, Subjects and Regions were taken into account:

*mod <- lm(gene expression ~ Sex + Age + Ancestry + PMI + Batch + BrainBank + RIN + seqCovs).*

This method allowed us to remove variation explained by biological and technical covariates.

Adjusted expression was calculated by extracting the residuals per each gene and adding the mean of the gene expression: *adjusted gene expression <− residuals(mod) + mean(gene expression)*

Adjusted CPM values were used for rs-fMRI—gene expression correlation and resultant visualization.

**fMRI-gene expression correlation analysis.** We performed Spearman's rank correlation between the mean regional values of fALFF and ReHo and the regional gene expression across the 11 cortical areas analyzed. To define fMRI-gene expression relationships, we used random subsampling (200 times) of neurotypical individuals from the ABIDE I and II datasets. We matched the number of subjects per each cortical area (e.g., 25 ASD subjects for BA17). We performed correlation across the regions using all 11 areas matching with the gene expression dataset and averaged Spearman's rank statistics over the 200 subsamples. P-values from Spearman's rank statistics were adjusted by Benjamini–Hochberg FDR. Differential Correlation analysis was performed comparing the resulting Rho from neurotypical individuals to individuals with ASD for each gene using the *psych* (v2.0.12) package in R. We combined the resultant Differential Correlation p-values and effect sizes using a Fischer's combination test in R. Significant results are reported at FDR < 0.05 for neurotypical individuals' statistics and P-value of combined differential correlation at $p < 0.01$.

**Leave-one-region out (LoRo) analysis.** We performed the same subsampling approach followed by differential correlation analysis as described above leaving one region out at the time. This method allowed us to determine the effect of each region in the resultant $z$ from the differential correlation analysis between healthy individuals and autistic individuals. Next, we calculated the contribution of each region based on a principal component analysis using the resultant $z$-values. We visualized resultant contributions in a multi-dimensional plot.

**Developmental analysis.** The identification of gene clusters with different developmental trajectories was performed on DC genes using all subjects except for individuals above 60 yr as they were represented only in the ASD group.

We applied residualization as previously described removing the age from the covariates.

*mod <− lm(gene expression ~ Sex + Ancestry + PMI + Batch + BrainBank + RIN + seqCovs).*

Then, we scaled gene expression and divided genes into three clusters according to the scaled expression values of healthy subjects only, using the *Kmeans* function from the *amap* (v0.8) package in R. We plotted the developmental trajectories using the loess regression and *ggplot2* (v3.3.2) package in R. To make loess regression computationally possible, 8000 data points were randomly sampled. Repeated samplings yielded very similar patterns. We made no adjustments for developmental time points and the $x$-axis directly represents the age of the subjects. We annotated clusters based on visual inspection of their trajectory. To subsample diagnosis-region groups (e.g., ASD BA17 samples), we determined the diagnosis-region group with minimum number of samples and randomly subset other groups to that number. Then we plotted expression values with loess regression as before. To assess the significance of trajectories, we compared gene expression between age brackets of 5 years using t-test (One-tailed. Greater expression for Adult (e.g Ha: 0–5 < 5–10) and less expression for EarlyDev (Ha: 0–5 > 5–10) with increasing age).

The BrainSpan dataset[48] was downloaded from www.brainspan.org (normalized matrix: "RNA-Seq Gencode v10 summarized to genes"). Data were then log2 transformed (log2(data + 1)). To match with the current study, the following brain regions were removed: AMY, OFC, Ocx, URL, DTH, CB, CBC, MD, STR, and HIP. For each gene, the expression values were z-transformed across samples. To understand expression pattern across ages, samples were divided into age groups per 5 years. Only postnatal samples were kept to match with the current study.

**Allen single nuclei RNA-seq analysis.** Multi-Region snRNA-seq[41] (MTG, V1C, M1C, CgGr, S1C, A1C) was from the Allen Brain Map portal (https://portal.brain-map.org/atlases-and-data/rnaseq). Briefly, data was analyzed using Seurat[94] (v3.9.9). Data was subsetted by removing nuclei with >10,000 UMI and >5% of mitochondrial gene expressed. Published cell-type annotations included in the metadata were used for downstream analyses. We identified cell-type markers using *FindMarkers* function based on Wilcoxon-rank sum test statistics. Markers were defined by Percentage of Cells expressing the gene in the cluster >0.5, FDR < 0.05 and $|log_2(FC)| > 0.3$.

**Functional enrichment**. We performed the functional annotation of differentially expressed and co-expressed genes using ToppGene[95]. We used the GO and KEGG databases. Pathways containing between 5 and 2000 genes were retained. We applied a Benjamini–Hochberg FDR ($P < 0.05$) as a multiple comparisons adjustment. Brain expressed genes (Brainspan, $N = 15585$) were used as background.

**Gene set enrichment**. We performed gene set enrichment for neuropsychiatric DGE[55], neuropsychiatric modules[55], and cell-type markers[41] using a Fisher's exact test in R with the following parameters: alternative = "greater", conf.level = 0.95. We reported odds ratios (OR) and Benjamini–Hochberg adjusted $P$-value (FDR). Brain expressed genes (Brainspan, $N = 15585$) were used as background.

**Deconvolution**. Deconvolution was performed by *MuSiC* (v0.1.1)[53] in R. This method leverages transcriptomic signatures of cell-types considering cross-subject heterogeneity and gene expression stochasticity. Bulk RNA-seq data is deconvoluted to obtain proportions of cell-types in each sample. We used single-cell data that was downloaded from the Allen Brain Map portal (https://portal.brain-map. org/atlases-and-data/rnaseq). Published cell-type annotations included in the metadata were used as reference for cell-type abundance inference.

**Statistical analysis and reproducibility**. No statistical methods were used to pre-determine sample sizes. Nevertheless, the data here reported is in line with the sample size of previous studies[96,97]. Samples were not randomized. ASD subjects with Chromosome 15q Duplication were excluded from the downstream analysis. Data collection and analysis were not performed blind to the conditions of the experiments. Findings were not replicated due to the limitation of the multi-region ASD transcriptome data. Nevertheless, we used two independent rs-fMRI measurement to refine and increase the confidence of our findings. For fALFF/ReHo rs-fMRI values and bulk RNA-seq transcriptomic data, distribution was assumed to be normal but this was not formally tested. Non-parametric tests have been used to avoid uncertainty when possible. Data collection and analysis were not performed blind to the conditions of the experiments.

**Reporting summary**. Further information on research design is available in the Nature Research Reporting Summary linked to this article.

## Data availability

The imaging data from ABIDE I and II are available to approved investigators who register with the NITRC (Neuroimaging Informatics Tools and Resources Clearinghouse) and 1000 Functional Connectomes Project to gain access. Details and access information are provided here: http://fcon_1000.projects.nitrc.org/indi/abide/abide_I.html and here: http://fcon_1000.projects.nitrc.org/indi/abide/abide_II.html.

The source bulk RNA-seq data generated in this manuscript are available via the PsychENCODE Knowledge Portal (https://psychencode.synapse.org/). The PsychENCODE Knowledge Portal is a platform for accessing data, analyses, and tools generated through grants funded by the National Institute of Mental Health (NIMH) PsychENCODE program. Data is available for general research use according to the following requirements for data access and data attribution: (https://psychencode. synapse.org/DataAccess). For access to content described in this manuscript see: https:// doi.org/10.7303/syn4587615.

## Code availability

Custom R code and data to support the data correction, correlation analysis, visualizations, functional, and gene set enrichments are available at https://github.com/konopkalab/ AUTISM_rsfMRI_GeneExpressionCorrelations and https://github.com/DeepLearning ForPrecisionHealthLab/AUTISM_rsfMRI_ProcessingConnectivityExtractionAndSubject Matching.

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

## Acknowledgements

G.K. is a Jon Heighten Scholar in Autism Research and Townsend Distinguished Chair in Research on Autism Spectrum Disorders at UT Southwestern Medical Center. E.C. is a Neural Scientist Training Program Fellow in the Peter O'Donnell Brain Institute at UT Southwestern. Data were generated as part of the PsychENCODE Consortium. Visit 10.7303/syn24240356 for a complete list of grants and PIs. Tissue specimens and/or data used in this research were obtained from the Autism BrainNet (formerly the Autism Tissue Program), which is sponsored by the Simons Foundation, and the University of Maryland Brain and Tissue Bank, which is a component of the NIH NeuroBiobank. We are grateful to the patients and families who participate in the tissue donation programs. Funding for this work was provided by grants to D.H.G. (NIMH R01MH110927, U01MH115746, P50-MH106438, and R01 MH-109912, R01 MH094714), grants to M.J.G. (SFARI Bridge to Independence Award, NIMH R01-MH121521, NIMH R01-MH123922, NICHD-P50-HD103557), and grants to J.R.H. (Achievement Rewards for College Scientists Foundation Los Angeles Founder Chapter, UCLA Neuroscience Interdepartmental Program). This work was also supported by the NIMH (MH102603, MH126481), NINDS (NS106447, NS115821), NHGRI (HG011641), the Simons Foundation (SFARI #573689), and the James S. McDonnell Foundation 21st Century Science Initiative in Understanding Human Cognition—Scholar Award (220020467) to G.K.

## Author contributions

S.B., E.C., A.H.T., D.L., A.A.M., and G.K. analyzed the data and wrote the paper. J.R.H., M.J.G., and D.H.G. collected samples, processed RNA, and generated bulk RNA-seq libraries. A.H.T. analyzed the ABIDE I and ABIDE II data. A.A.M. and G.K. designed and supervised the study, and provided intellectual guidance. All authors discussed the results and commented on the manuscript.

## Competing interests

The authors declare no competing interests.
