## [Peer Review File · Nature Communications]

Association between resting-state functional brain connectivity and gene expression is altered in autism spectrum disorderReviewers' comments:

Reviewer #1 (Remarks to the Author):

This is a strong paper that I think warrants publication in Nature Communications pending major revisions. The paper is well positioned to provide interesting cross-level insight into BOLD signal correlates of ASD. I am enthusiastic about the topic, data, and approach, however I feel like the methods and analyses require more precision and detailed description. It should be highlighted that this paper stands out among the recent trend linking neuroimaging markers of clinical disorders to postmortem patterns of gene expression. The bulk of these studies necessarily rely on non-clinical post-mortem data like BrainSpan or Allen Human Brain Atlas, limiting the strength of conclusions that can be drawn. This paper has a unique combination of spatial coverage (n=11 brain regions) and clinical sampling (ASD vs controls) in the RNAseq data. The choice of fALFF and ReHo as clinical phenotypes are also sensible given prior ASD literature and the topic is of interest and importance to the field. Although I like this paper, I think the neuroimaging analyses omit key preprocessing steps and insufficiently account for confounding sources of noise, like head motion. The bioinformatics also sometimes trade breadth for depth. I commend the authors for this nice cross-level work though and acknowledge the difficulty inherent to linking disparate data types. My comments are meant to be constructive and point out parts of the methods that require clarification or potential changes.

Side note to authors: I would love to see this analysis done using structural data – the case/control effect sizes for thickness and volume are not bad in ASD (<https://doi.org/10.1176/appi.ajp.2017.17010100>) and structural data avoids many pitfalls inherent to clinical functional neuroimaging (treat this statement about as an enthusiasm for future work, not a request for inclusion of structural data in this paper).

Comments:

1. The last paragraph of the introduction reads like a results section. It would strengthen the paper to instead discuss broader questions about why the work is important and which gaps in the literature are being filled.
2. The neuroimaging analyses diverge from field norms and need to more explicitly address the potential influence of head motion, which is one of the most important sources of confounding for functional imaging studies of autism.
 - a. Sensitivity analyses are required to test for the effects of head motion. Did ASD/CTL groups differ in terms of DVARS and avg frame-wise displacement (FWD)? Do the results hold up if only low-motion ASD subjects are analyzed? This is particularly important since head motion can induce structure functional artifacts that appear as gradients. Last, T1 SNR, rfMRI SNR, and avg DVARS+FWD should be likely be covaried in the analyses.
 - b. There is no mention of how time series were residualized for head motion parameters (X/Y/Z/pitch/roll/yaw + their derivatives). This is a huge issue and analyses must be rerun if this step was

omitted.

c. BOLD timeseries were residualized for global signal, but why wasn't the average signal from white matter and ventricles included as well?

d. Similar to above, were DVARS and FWD outliers censored? This is a recommended preprocessing step. See this nice breakdown about why this is important, and how to order censoring/interpolation/bandpass steps done in the correct order, as in Power 2014; 10.1016/j.neuroimage.2013.08.048 (https://github.com/ThomasYeoLab/CBIG/blob/master/stable_projects/preprocessing/CBIG_fmRI_Preproc2016/Recommendation_of_bandpass_censoring.md)

e. The authors define average fALFF/ReHo maps in ASD and CTL groups (from residualized site-binned data). The spatial configuration of these maps were then correlated to the gene expression. Would it be more straightforward to conduct a linear mixed effects (LME) model to explicitly identify ASD/CTL group differences and conduct a single correlation test between gene expression and the fMRI group difference map? (Brodman area Cohen's d or equivalent). Alternatively, instead of an LME approach, the authors could meta-analyze imaging data from each site, similar to how the ENIGMA consortium analyses site data. Currently, I have a good sense for how fALFF/ReHo differs between ASDs and neurotypical individuals.

f. I think visual brain plots of fALFF/ReHo in ASD and CTLs would go a long way. Also moving Supp Fig. 3b to main results. The most useful plot though would show statistical differences in ReHo/fALFF between ASDs and CTLs.

g. What was the basis for determining image artifacts and MNI registration failure?

h. Given the heterogeneity in scan acquisitions, I'm a bit worried about the lack of detail about important analysis steps that would cause quite a headache to get correct across sites. Were initial EPI frames censored from BOLD runs to allow for field stabilization? Was slice time-correction conducted and are the authors confident that the slice interleaving was correctly specified?

i. There is also no mention of censoring subjects based on overall head motion. For instance, it's common to remove subjects based on a sensible motion threshold, like average FWD > 0.30mm or greater than 50% of frames being censored.

j. The authors should report T1 and rest run acquisition parameters (possible in supplemental table). This information should be summarized a bit in the methods however to give the reader a sense of how variable were sites in terms of scanner, head coils, scan length, scan resolution, etc?

k. Was there a uniform instruction for subjects to keep eyes open or closed during REST runs? This influences fALFF and is important given the highlighted effects in visual cortex. Given differences in ASD visual processing, groups could differ in terms of eyes open/closed. I'm not expecting a thorough answer to this since ABIDE is a collaborative open dataset, but I think it should be acknowledged.

3. Might help with readability to assign descriptive labels to brodmann areas. E.g. "BA20_37" becomes "BA20_37 (Ventral Temporal)".
4. The authors might find the dataset from Krienen et al 2020 of use for future work (<https://www.nature.com/articles/s41586-020-2781-z.pdf?origin=ppub>).
5. More information is required upfront in the "Differentially correlated genes have specific developmental trajectories" section. Which dataset was analyzed (BrainVar/Brainspan?) dev dataset was analyzed, this section. How was between subject's normalization conducted (e.g. TMM in DESeq)? Were low-expressed genes removed? Details like this should be included.
6. Related to the above point, individual data points should be plotted for Figure 3A. There is usually heterogeneous sampling across the age range that readers should be oriented toward.
7. The bubble charts in Figure 3b are tough to read. Were there only three enrichment terms for adult/earlydev/stable gene bins? Also the lines from each circle were confusing. I wasn't sure if they were a feature of the data plot or if they were linking the circles to a corresponding label. I realized it's the latter, but I would make this uniform and add lines between each dot and each to make it less confusing.
8. Fig 2c is unclear to me. What variance is being explained? What does each dot represent?
9. The authors highlight PVALB gene gradients, which is consistent with the literature. But it is increasingly acknowledged that the PVALB posterior-to-anterior expression pattern is non-specific and part of a larger gene gradient that includes markers of other cell types. How does the fALFF/PVALB correlation compare to that of first and second principle components of gene expression?
10. The potential neurovascular components of fALFF and BOLD signal amplitude should be discussed
11. Was the linear and nonlinear transform to MNI space combined and conducted as a single step to reduce distortion?
12. Apologize if I'm being dense, but I can't find Table 1.
13. Any differences in frozen or fresh tissue in the differential expression data? There is also no information about how RNAseq data were processed (e.g. STAR? RSEM?) or which genome assembly was used.
14. For functional enrichment analyses, what was the background set?
15. Very little information about the protein-protein interaction analysis. What is a string score? How should the data be interpreted?

16. There is also almost no information about the deconvolution analyses. This is a very tricky analysis to pull off and depends heavily on the granularity of single-cell cell grouping. Collinearity can be a big issue, for instance, if you try to deconvolve highly similar neuronal classes.

17. There is also little info on the single-cell data or analyses.

Minor comments:

1. "We collected the ABIDE data", should likely say "we accessed publicly available ABIDE data"

2. Spacing formatting of picard covariate list.

Reviewer #2 (Remarks to the Author):

This study reports gene-brain activity links that are disrupted in individuals with autism. A subset of genes (enriched in voltage-gated ion channels and inhibitory neurons) showed differential developmental trajectories in autism. Primary visual cortex was found to be the most affected brain region in autism. Overall, there is no clear rationale for any of the analytic decisions made, making the contribution of these findings to the autism literature quite limited.

The motivation for the study is not clear from the introduction. There is not enough background describing the potential mechanisms whereby gene expression influences the development and maintenance of resting state functional brain networks.

On page 2 the authors write "We computed two extensively validation measures of brain activation..." fALFF is not a measure of brain activation. It is derived from resting-state fMRI data, where participants are not instructed to perform a cognitive task. Similarly, ReHo is not a measure of brain activation, but rather of functional connectivity.

There is no rationale for why fALFF and ReHo data were analyzed, rather than other functional connectivity metrics commonly used in the resting state fMRI, network neuroscience, and connectomics literatures. Further, there is no theoretical basis for the assumption that there should be convergence between fALFF and ReHo.

There is not enough information provided regarding the resting state fMRI data analysis. Basic analytic decisions (eg. what were the head motion criteria/cutoffs) are not included.

Resting state fMRI and gene expression data were not available from the same subjects, limiting the interpretability of the presented results.

Reviewer #3 (Remarks to the Author):

This manuscript is a detailed study integrating single cell gene expression data sets from neurotypical control and autistic patients (AUD) with regionally matched brain activity measurements obtained from fMRI datasets. The authors identify genes linked with brain activity that is disrupted in AUD patients. The gene sets are found to be enriched in voltage-gated ion channels and inhibitory neurons. An interesting result is that of the regions profiled primary visual cortex is seen to be the most affected region in AUD patients. The use of control and patient specific transcriptome profiles is unique in this study and is an important strength.

This work represents continued investigation in determining molecular correlates of functional imaging results, studies that have been of interest to the neuroimaging community for some time and with increased feasibility as brain wide deeper profiled molecular data sets have become available. The problem is important but challenging for several reasons including small effect sizes and highly correlated gene sets. This study has some of the same challenges but the approach and methods in the study are sound and well analyzed comparing disparate data sets. The authors have taken a reasonably comprehensive approach to setting up the problem and analysis. Some comments for consideration are:

1. Regions exhibiting significance differences in fALFF and ReHO are seen to be significant at a fairly weak level ($p < 0.05$). Can one be more specific about the distribution of these effects, perhaps even with exemplars compared across regions profiled?
2. There are a very small number of genes found intersecting with previous studies such as Ricardi et al, and these are well known genes implicated with a variety of functions, e.g. PVALB, SCN1B, SYT2. How do the authors understand this limited intersection?
3. In examining the relationship to development, it would be interesting to compare with the BrainSpan (www.brainspan.org) dataset which, although I believe is not single cell, contains a reasonably wide developmental trajectory and might provide insights connecting this work with previous studies.
4. The differential correlation concept comparing control versus AUD is a strength of this study, and Fig 2. Illustrates the concept. It would be helpful to have some of the distribution properties of the associations found brought forward more transparently, perhaps a figure with ranking of genes by effect size.
5. The ontological and protein association studies should be perhaps controlled with respect to background, at least for comparable brain function datasets. Generally network presentations such as those of Fig 3c are not particularly revealing unless substantiated with further evidence

We would like to thank the editor and reviewers for their thoughtful critiques of our manuscript. We have completely redone all of our analyses and believe that we were able to address all comments.

Reviewers'

comments:

Reviewer #1 (Remarks to the Author):

This is a strong paper that I think warrants publication in Nature Communications pending major revisions. The paper is well positioned to provide interesting cross-level insight into BOLD signal correlates of ASD. I am enthusiastic about the topic, data, and approach, however I feel like the methods and analyses require more precision and detailed description. It should be highlighted that this paper stands out among the recent trend linking neuroimaging markers of clinical disorders to postmortem patterns of gene expression. The bulk of these studies necessarily rely on non-clinical post-mortem data like BrainSpan or Allen Human Brain Atlas, limiting the strength of conclusions that can be drawn. This paper has a unique combination of spatial coverage (n=11 brain regions) and clinical sampling (ASD vs controls) in the RNAseq data. The choice of fALFF and ReHo as clinical phenotypes are also sensible given prior ASD literature and the topic is of interest and importance to the field. Although I like this paper, I think the neuroimaging analyses omit key preprocessing steps and insufficiently account for confounding sources of noise, like head motion. The bioinformatics also sometimes trade breadth for depth. I commend the authors for this nice cross-level work though and acknowledge the difficulty inherent to linking disparate data types. My comments are meant to be constructive and point out parts of the methods that require clarification or potential changes.

Side note to authors: I would love to see this analysis done using structural data – the case/control effect sizes for thickness and volume are not bad in ASD (<https://doi.org/10.1176/appi.ajp.2017.17010100>) and structural data avoids many pitfalls inherent to clinical functional neuroimaging (treat this statement about as an enthusiasm for future work, not a request for inclusion of structural data in this paper).

We thank the reviewer for this positive feedback. We now amended the manuscript to answer all the concerns.

Comments:

1. The last paragraph of the introduction reads like a results section. It would strengthen the paper to instead discuss broader questions about why the work is important and which gaps in the literature are being filled.

Thank you for this suggestion. We have now substantially expanded the introduction.

2. The neuroimaging analyses diverge from field norms and need to more explicitly address the potential influence of head motion, which is one of the most important sources of confounding for functional imaging studies of autism.

We had previously addressed head motion but did not provide sufficient detail. In the revised manuscript, we now explicitly address head motion with additional details in the Methods:

“For subjects with multiple fMRI scans, the scan with the lowest head motion, measured by mean framewise displacement (FWD), is selected for analysis. For each resulting subject scan, a subject is excluded if their scan has excessive head motion. Specifically, scans meeting at least one of these 3 requirements are removed: (1) mean FWD > 0.30mm, (2) greater than 50% of frames being scrubbed, or (3) scans with outlier mean, 1st, 2nd, or 3rd quantile DVARS values. DVARS is defined as the root mean square of the temporal change of the fMRI voxel-wise signal at each time point^{82,83}. The package CPAC v1.8.0 is used for fMRI pre-processing including head motion correction, scrubbing, and nuisance regression.”

a. Sensitivity analyses are required to test for the effects of head motion. Did ASD/CTL groups differ in terms of DVARS and avg frame-wise displacement (FWD)? Do the results hold up if only low-motion ASD subjects are analyzed? This is particularly important since head motion can induce structure functional artifacts that appear as gradients. Last, T1 SNR, rfMRI SNR, and avg DVARS+FWD should be likely be covaried in the analyses.

We rigorously culled subjects with excessive head motion (as detailed in section “fMRI preprocessing”) in order to minimize potential for motion to influence our findings. The results of our analysis therefore apply when the low motion ASD subjects are analyzed.

FWD is explicitly regressed out and we exclude subjects with excessive FWD. We exclude subjects with high (excessive) DVARS, which minimizes variation in DVARS. We regress out the global signal in rsfMRI. Regressing out T1 SNR, rsfMRI SNR are currently non-standard approaches and we did not apply them for several reasons. First, there is a lack of consensus in the community as evidenced by three of the most widely used preprocessing pipelines [1,2,3] not providing such covariate regression. Second, the standard preprocessed ABIDE public dataset [4] does not apply them. Finally, we note that T1 is a separately acquired MRI contrast whose covariate regression could introduce additional confounds which we wish to avoid.

1. CPAC: <https://fcp-indi.github.io/docs/latest/user/nuisance>
2. fMRI prep: <https://fmriprep.org/en/stable/outputs.html>
3. Conn: https://web.conn-toolbox.org/fmri-methods/denoising-pipeline#h.p_m3LmQHcwakjM
4. Preprocessed ABIDE (CPAC): <http://preprocessed-connectomes-project.org/abide/cpac.html>

b. There is no mention of how time series were residualized for head motion parameters (X/Y/Z/pitch/roll/yaw + their derivatives). This is a huge issue and analyses must be rerun if this step was omitted.

We addressed these issues and include this information in the Methods:

“Images are processed with a generalized linear model (GLM) to regress out: 1) global signal fluctuation, 2) physiological noise represented by white matter and CSF fluctuation, 3) fluctuation correlated with the 6 original affine head motion parameters (X/Y/Z/pitch/roll/yaw), 4) their first derivatives, squares, and squared derivatives, and 5) noise fluctuations captured from 5 components from aCompCor⁸⁰.”

c. BOLD timeseries were residualized for global signal, but why wasn’t the average signal from white matter and ventricles included as well?

We thank the reviewer for this comment. We now include signal for white matter and the CSF filled ventricles in our GLM, as stated in response to the previous comment (2b) above.

d. Similar to above, were DVARS and FWD outliers censored? This is a recommended preprocessing step. See this nice breakdown about why this is important, and how to order censoring/interpolation/bandpass steps done in the correct order, as in Power 2014; 10.1016/j.neuroimage.2013.08.048 (https://github.com/ThomasYeoLab/CBIG/blob/master/stable_projects/preprocessing/CBIG_fMRI_Preproc2016/Recommendation_of_bandpass_censoring.md)

We thank the review for this question. We now filter based on this suggestion as stated in the methods:

“Specifically, scans meeting at least one of these 3 requirements are removed: (1) mean FWD > 0.30mm, (2) greater than 50% of frames being scrubbed, or (3) scans with outlier mean, 1st, 2nd, or 3rd quantile DVARS values.”

e. The authors define average fALFF/ReHo maps in ASD and CTL groups (from residualized site-binned data). The spatial configuration of these maps were then correlated to the gene expression. Would it be more straightforward to conduct a linear mixed effects (LME) model to explicitly identify ASD/CTL group differences and conduct a single correlation test between gene expression and the fMRI group difference map? (Brodman area Cohen’s d or equivalent). Alternatively, instead of an LME approach, the authors could meta-analyze imaging data from each site, similar to how the ENIGMA consortium analyses site data. Currently, I have a good sense for how fALFF/ReHo differs between ASDs and neurotypical individuals.

f. I think visual brain plots of fALFF/ReHo in ASD and CTLs would go a long way. Also moving Supp Fig. 3b to main results. The most useful plot though would show statistical differences in ReHo/fALFF between ASDs and CTLs.

We thank the reviewer for both of these comments (e. and f.). We now included a brain visualization as figure 2 depicting the differences between ASD and CTL in the ROI analyzed (Cohen's d). We also included a scatter plot depicting the spatial correlation between Cohen's d calculated by the two different measurements.

Fig. 2: Imaging differences between ASD and CTL. a, Differences between ASD and CTL calculated by Cohen's d (effect sizes) derived from ASD – CTL comparison for both rs-fMRI measurements across the ROIs analyzed. b, Scatter plot depicting the spatial correlation between Cohen's d values of fALFF and ReHo. Each dot corresponds to the ROI analyzed.

We amended the text to reflect these changes

“We first assessed differences between cases and controls for both fALFF and ReHo (Fig. 2a). We identified 4 ROIs with a significant difference for fALFF and 1 ROIs for ReHo (Wilcoxon Rank Sum Test, $p < 0.05$; Supplementary Fig. 2a). BA20/37 was commonly different using either measurement. Even though we observed small effect sizes for all the ROIs analyzed (Cohen's d; $d < 0.3$) in agreement with other reports, we observed consistency between fALFF and ReHo (Spearman Rank Correlation, $\rho = 0.46$; Fig. 2b). These data reflect subtle, yet replicable functional activity measurements linked to ASD calculated by two rs-fMRI measurements.”

g. What was the basis for determining image artifacts and MNI registration failure?

We removed subjects with image artifacts, high head movement, or inadequate MNI152 coregistration. Determination of residual physiological and motion artifacts is achieved through a combination of scrubbing, aCompCor, and global signal regression.

We amended the Registration section on page 24 of the manuscript to clarify this:

“Lastly subjects with poor EPI_{norm} registration⁹⁰ (discussed below) are removed. Specifically, mis-registration is identified through a combination of manual inspection and through the detection of scans with an outlier number of misaligned brain-masked voxels using the interquartile range (IQR) outlier test⁹¹.”

h. Given the heterogeneity in scan acquisitions, I'm a bit worried about the lack of detail about important analysis steps that would cause quite a headache to get correct across sites. Were initial EPI frames censored from BOLD runs to allow for field stabilization? Was slice time-correction conducted and are the authors confident that the slice interleaving was correctly specified?

We now include additional data to address slice correction:

“The first 5 volumes are censored to allow for MRI scanner dynamic instability to settle. To correct for head movement, volume realignment was applied frame by frame, to register each volume to the mean volume with an affine transformation. Slice timing correction is applied to ensure volume slices align temporally.”

i. There is also no mention of censoring subjects based on overall head motion. For instance, it's common to remove subjects based on a sensible motion threshold, like average FWD > 0.30mm or greater than 50% of frames being censored.

We now address this issue in Methods:

“Specifically, scans meeting at least one of these 3 requirements are removed: (1) mean FWD > 0.30mm, (2) greater than 50% of frames being scrubbed, or (3) scans with outlier mean, 1st, 2nd, or 3rd quantile DVARS values.”

j. The authors should report T1 and rest run acquisition parameters (possible in supplemental table). This information should be summarized a bit in the methods however to give the reader a sense of how variable were sites in terms of scanner, head coils, scan length, scan resolution, etc?

We now include the table (shown below) within Supplementary table 1.

ABIDE	SITE	scanner manufacturer	head coil channels	T1 scan resolution (mm)	fMRI scan length (min:sec)	fMRI scan resolution (mm)	fMRI TR (ms)
1	CALTECH	Siemens	8	1.0x1.0x1.0	5:04	3.5x3.5x3.5	2000
1	CMU	Siemens	8	1.0x1.0x1.0	8:06	3.0x3.0x3.0	2000
1	KKI	Phillips	8	1.0x1.0x1.0	4:40	3.05x3.15x3	2500
1	LEUVEN_1	Phillips	8	.975x.975x1.2	7:00	3.59x3.59x4	1667
1	LEUVEN_2	Phillips	8	.975x.975x1.2	8:00	3.59x3.59x5	1668
1	MAX_MUN	Siemens	8	1.0x1.0x1.0	6:06	3.0x3.0x4.0	3000
1	NYU	Siemens	unavailable	1.3x1.0x1.3	6:00	3.0x3.0x4.0	2000
1	OHSU	Siemens	8	1.0x1.0x1.1	3:32	3.8x3.8x3.8	2500
1	OLIN	Siemens	unavailable	1.0x1.0x1.0	5:15	3.4x3.4x4.0	1500
1	PITT	Siemens	unavailable	1.1x1.1x1.1	5:06	3.1x3.1x4.0	1500
1	SBL	Philips	32	1.0x1.0x1.0	7:28	2.75x2.75x2.72	2200
1	UCLA_1	Siemens	8	1.0x1.0x1.2	6:06	3.0x3.0x4.0	3000
1	UCLA_2	Siemens	8	1.0x1.0x1.2	6:06	3.0x3.0x4.0	3000
1	UM_1	GE	4	1.0x1.0x1.4	10:00	3.438x3.438x3	2000
1	UM_2	GE	4	1.0x1.0x1.4	10:00	3.438x3.438x3	2000
1	USM	Siemens	8	1.0x1.0x1.2	8:06	3.4x3.4x3.0	2000
1	YALE	Siemens	8	1.0x1.0x1.0	6:40	3.4x3.4x4.0	2000
2	BNI_1	Philips	15	1.06x1.06x1.0	6:09	3.75x3.75x4	3000
2	EMC_1	GE	8	0.9x0.9x0.9	8:20	3.6x3.6x4.0	2000
2	ETH_1	Philips	32	0.9x0.9x0.9	7:06	3.0x3.1x3.0	2000
2	IU_1	Siemens	32	0.7x0.7x0.7	16:21	3.4x3.4x3.4	813
2	KKI_1(8 ch)	Philips	8	1.0x1.0x1.0	6:40	3.05x3.15x3	2500
2	KKI_1(32 ch)	Philips	32	0.95x0.95x1.0	6:40	3.05x3.15x3	2500
2	NYU_1	Siemens	8	1.3x1.0x1.33	6:00	3.0x3.0x4.0	2000
2	OHSU_1	Siemens	12	1.0x1.0x1.1	5:07	3.8x3.8x3.8	2500
2	SDSU_1	GE	8	1.0x1.0x1.0	6:10	3.4375x3.4375x	2000
2	TCD_1	Philips	8	0.9x0.9x0.9	7:06	3.0x3.0x3.2	2000
2	UCD_1	Siemens	32	1.0x1.0x1.0	5:06	3.5x3.5x3.5	2000
2	UCLA_1	Siemens	12	1.0x1.0x1.2	6:06	3.0x3.0x4.0	3000
2	USM_1	Siemens	12	1.0x1.0x1.2	8:06	3.4x3.4x3.0	2000

k. Was there a uniform instruction for subjects to keep eyes open or closed during REST runs? This influences fALFF and is important given the highlighted effects in visual cortex. Given differences in ASD visual processing, groups could differ in terms of eyes open/closed. I'm not expecting a thorough answer to this since ABIDE is a collaborative open dataset, but I think it should be acknowledged.

As the reviewer mentions, we are unable to sufficiently answer this due to the nature of the public dataset. We now include this possibility as a caveat in the discussion:

“Because subjects who underwent fMRI measurements might not have had uniform instructions (or resultant behavioral compliance) to keep their eyes open or closed, it is possible that the visual cortex data could be influenced by such behavior.”

3. Might help with readability to assign descriptive labels to Brodmann areas. E.g. “BA20_37” becomes “BA20_37 (Ventral Temporal)”.

Thank you for this suggestion. We have now included descriptive labels when we first note the ROIs in the text.

“...using Brodmann area (BA) designations: BA1/2/3/5 (somatosensory cortex), BA4/6 (premotor and primary motor cortex), BA7 (superior parietal gyrus), BA9 (dorsolateral prefrontal cortex), BA17 (primary visual cortex), BA20/37 (inferior temporal cortex), BA24 (dorsal anterior cingulate cortex), BA38 (temporal pole), BA39/40 (inferior parietal cortex), BA41/42/22 (superior temporal gyrus), BA44/45 (inferior frontal gyrus).”

4. The authors might find the dataset from Krienen et al 2020 of use for future work (<https://www.nature.com/articles/s41586-020-2781-z.pdf?origin=ppub>).

We thank the reviewer for the suggestion. In future studies that include imaging data from striatum it would be quite interesting to investigate any potential role for primate innovations in interneurons and their potential contribution to gene expression patterns that underlie human brain activity and/or disease.

5. More information is required upfront in the “Differentially correlated genes have specific developmental trajectories” section. Which dataset was analyzed (BrainVar/BrainSpan?). How was between subject’s normalization conducted (e.g. TMM in DESeq)? Were low-expressed genes removed? Details like this should be included.

We thank the reviewer for this comment. The dataset we utilized for rs-fMRI correlation is collected across ages (2-60 yrs old) for both healthy controls (CTL) and ASD patients. Therefore, we simply made use of our dataset to unravel the developmental patterns of DC (differentially correlated) genes both in CTL and ASD. We now state this more explicitly:

*“We leveraged **the transcriptomic dataset from this study** to detect whether DC genes follow a specific developmental trajectory in ASD compared with CTL subjects (see Methods).”*

We now additionally analyzed the BrainSpan dataset upon the suggestion from reviewer #3, which confirms our findings for the CTL group (Supplementary Figure 6c). Both datasets were normalized and log transformed. We then only analyzed DC genes which were not lowly expressed in either dataset. Regarding between subject variation, we removed effects explained by covariates for this study’s dataset as described in the methods. BrainSpan covariates were not removed as they were either relevant information (e.g. brain region, age) or not documented in the original study. Given that the two different datasets agree on the developmental trajectory, we also do not think between subject variation contributes substantially to this analysis.

6. Related to the above point, individual data points should be plotted for Figure 3A. There is usually heterogeneous sampling across the age range that readers should be oriented toward.

The reviewer raises an important point. We used loess regression, which does smoothing that is useful for highlighting the overall pattern but masks the underlying heterogeneity. To be more explicit about heterogeneity within and across age groups, we now also split ages into groups of 5 years, created boxplots per age group and statistically compared the age groups (Supplementary Figure 6a). We prefer to provide the boxplots instead of the plots of each data point as the data points are too many and it is difficult to interpret (see Revision Fig 1 below).

Supplementary Figure 6. a, Statistical comparison of developmental trajectories. Samples were divided into age brackets and age brackets were compared by one-sided t-test (alternative hypothesis: greater with increasing age in Adult, less with increasing age in EarlyDev). Numbers on graph are p-values, y-axis indicates Z-scored gene expression. Note that z-score spans a larger interval compared to Figure 4a.

Revision Figure 1: Developmental trajectory overlaid on data points. Each data point represents an expression value of a gene for the given sample.

7. The bubble charts in Figure 3b are tough to read. Were there only three enrichment terms for adult/earlydev/stable gene bins? Also the lines from each circle were confusing. I wasn't sure if they were a feature of the data plot or if they were linking the circles to a corresponding label. I realized it's the latter, but I would make this uniform and add lines between each dot and each to make it less confusing.

We thank the reviewer for the suggestion. We updated the figure based on the new data which changed the format of the bubble chart. There are indeed more significantly enriched categories than we plotted in the figure (now Figure 4b). We plotted only a selection of top category enrichments. The full list of enrichment categories can now be found in Table S2.

8. Fig 2c is unclear to me. What variance is being explained? What does each dot represent?

We thank the reviewer for the comment. Because this figure is no longer included in the revised manuscript, we removed the relevant text and figure legend.

9. The authors highlight PVALB gene gradients, which is consistent with the literature. But it is increasingly acknowledged that the PVALB posterior-to-anterior expression pattern is non-specific and part of a larger gene gradient that includes markers of other cell types. How does the fALFF/PVALB correlation compare to that of first and second principle components of gene expression?

We thank the reviewer for the comment. We followed the suggestion of the reviewer and analyzed the coupling between PC1 and PC2 of gene expression with PVALB and SCN1B gene expression. Using a principal component analysis, we found a similar pattern between expression (CTL), rs-fMRI measurements, and PVALB and SCN1B gene expression. We now include this figure as Figure 3g and we amended the text accordingly.

Fig. 3g: Gradient of CTL expressions (PC1), PVALB, and SCN1B gene expression. Barplot depicts correlation between PVALB and SCN1B gene expression with ReHo, fALFF, expression PC1 and expression PC2.

10. The potential neurovascular components of fALFF and BOLD signal amplitude should be discussed

Psychological noise, including neurovascular dependent noise, has been shown to confound fMRI [PMIDs 11590638, 16488843]. However, aCompCor [PMID 17560126] is an effective strategy to help mitigate these effects on resting state fMRI and computation of derivatives such as fALFF. To minimize confounds from psychological noise in our analysis, aCompCor is included in the nuisance regression.

11. Was the linear and nonlinear transform to MNI space combined and conducted as a single step to reduce distortion?

Yes, the linear and non-linear transform were combined and conducted as a single step. The combined transform was applied to the derivative maps during the EPI registration to the MNI template.

12. Apologize if I'm being dense, but I can't find Table 1.

We do apologize for this inconvenience. The Supplementary Table 1 is now available.

13. Any differences in frozen or fresh tissue in the differential expression data? There is also no information about how RNAseq data were processed (e.g. STAR? RSEM?) or which genome assembly was used.

We thank the reviewer for this comment. All of these data were derived from frozen post-mortem tissue. The data used in this project was shared with us and published in this preprint (<https://www.biorxiv.org/content/10.1101/2020.12.17.423129v1.full>). We worked closely with the Geschwind and Gandal groups, and we processed the data ourselves to match their preprocessing. We have now amended the methods text to include additional information on the preprocessing of the data:

“Quality control was performed using FastQC (v.0.11.9). Reads were aligned to the human hg38 reference genome using STAR⁹² (v.2.5.2b). Picard tool was implemented to refine the quality control metrics (<http://broadinstitute.github.io/picard/>) and to calculate sequencing statistics. Gencode annotation for hg38 (v.25) was used for reference alignment annotation and downstream quantification. Gene level expression was calculated using RSEM⁹³.”

14. For functional enrichment analyses, what was the background set?

We thank the reviewer for the comment. We used brain expressed genes obtained from the BrainSpan dataset (N = 15585). We amended the method text to reflect this information.

15. Very little information about the protein-protein interaction analysis. What is a string score? How should the data be interpreted?

We thank the reviewer for the comment. Based on this comment and that of reviewer 3, we decided to remove the PPI interaction from the current version of the manuscript.

16. There is also almost no information about the deconvolution analyses. This is a very tricky analysis to pull off and depends heavily on the granularity of single-cell cell grouping. Collinearity can be a big issue, for instance, if you try to deconvolve highly similar neuronal classes.

We thank the reviewer for the comment. We acknowledge that deconvolution may not yield robust results for highly similar cell types. In our case, this problem is alleviated by several details. First, we only focus on PVALB interneurons, which have relatively distinct profiles even among the inhibitory neuron populations (Supplementary Figure 6d). Second, our reference single-cell dataset is a NeuN+ enriched Smart-Seq2 dataset which is especially powered to detect changes among the transcriptional profiles of neurons. Third, we only deconvolute using a select number of genes and some of them are already enriched for specific neuronal types; for example, Adult cluster is highly enriched in PVALB cell type markers (Fig 4c).

We also now amended the method text including additional information about the deconvolution analysis.

“Deconvolution was performed by MuSiC (v0.1.1)⁵³ in R. This method leverages transcriptomic signatures of cell-types considering cross-subject heterogeneity and gene expression stochasticity. Bulk RNA-seq data is deconvoluted to obtain proportions of cell-types in each sample. We used single-cell data that was downloaded from the Allen Brain Map portal (<https://portal.brain-map.org/atlasses-and-data/maseq>). Published cell-type annotations included in the metadata were used as reference for cell-type abundance inference.”

17. There is also little info on the single-cell data or analyses.

We thank the reviewer for the comment. The data is publicly available in the Allen Brain Institute website and it has been preprocessed by them. Nevertheless, we now amended the method text including additional information about the single cell data analysis.

Minor comments:

1. “We collected the ABIDE data”, should likely say “we accessed publicly available ABIDE data”

Thank you. We now corrected the sentence.

2. Spacing formatting of picard covariate list.

Thank you. We now corrected the spacing.

Reviewer #2 (Remarks to the Author):

This study reports gene-brain activity links that are disrupted in individuals with autism. A subset of genes (enriched in voltage-gated ion channels and inhibitory neurons) showed differential developmental trajectories in autism. Primary visual cortex was found to be the most affected brain region in autism. Overall, there is no clear rationale for any of the analytic decisions made, making the contribution of these findings to the autism literature quite limited.

The motivation for the study is not clear from the introduction. There is not enough background describing the potential mechanisms whereby gene expression influences the development and maintenance of resting state functional brain networks.

We apologize that the background was insufficient and did not clearly address our rationale and motivation. We have now expanded the introduction in the revised manuscript. However, we would like to note that to understand potential gene regulatory mechanisms, the first task is to identify reliable sets of genes that correlate with functional brain network measurements. This was essentially the motivation that marked the start of the brain imaging – genomics field which our group also pioneered (PMIDs: 26590343, 26068849). While mechanistic insight needs to be studied by functional experiments, there are a number of recent publications – including in Nature Communications – that took a similar approach to identify genes spatially correlated with resting state brain networks and potentially disrupted in psychiatric diseases (PMIDs: 33077750, 30482947). We believe such studies are important in prioritizing genes for functional studies on susceptible genes which will be low-throughput and time expensive.

On page 2 the authors write “We computed two extensively validated measures of brain activation...” fALFF is not a measure of brain activation. It is derived from resting-state fMRI data, where participants are not instructed to perform a cognitive task. Similarly, ReHo is not a measure of brain activation, but rather of functional connectivity.

We respectfully disagree (and reviewers #1 and 3 thought our approach was sound in this regard). The brain is highly active when not performing a specific task. Moreover, when asked to do a specific cognitive task the entire brain doesn't switch to that task, only a small part of the signal changes. 80% of the variance in a fMRI task is still resting state signal (PMID: 24991964, PMID: 30708106). fALFF measures the amplitude of fluctuation and is a measure of a subset of brain activity within the low frequency band, and that activity is vitally important whether at rest (daydreaming, musing) or attending to a specific task as also outlined in C-PAC documentation (<https://fcp-indi.github.io/docs/latest/user/alff.html?highlight=falff>)

ReHo is a measure of the similarity of brain activity at a voxel to the voxels near it. As it is a derived property of brain activity, we technically agree with the reviewer that it is a measure of local functional connectivity, but that measure is itself a close derivative of the underlying brain activity. Citing the C-PAC documentation: “Regional Homogeneity (ReHo) is a voxel-based measure of brain activity which evaluates the similarity or synchronization between the time series of a given voxel and its nearest neighbors (Zang et al., 2004).” <https://fcp-indi.github.io/docs/latest/user/reho>

There is no rationale for why fALFF and ReHo data were analyzed, rather than other functional connectivity metrics commonly used in the resting state fMRI, network neuroscience, and connectomics literatures. Further, there is no theoretical basis for the assumption that there should be convergence between fALFF and ReHo.

We thank the reviewer for this question and acknowledge that our rationale needed more detail in the original submission. We selected local measures of resting state fMRI in order to match gene expression and fMRI measures for each brain region. We could, in principle, use functional connectivity metrics but this would derive information from brain regions not represented by our gene expression dataset. Therefore, we purposefully chose fALFF and ReHo as they are some of the only local measures of resting state fMRI.

We are not unfamiliar with ReHo and fALFF. We just published their very similar results for predicting the current severity (diagnosis) and future severity (1,2,4 years into the future) for Parkinson's Disease (PMID: 33730626). They explain similar amounts of variance in this regression challenge. This is not too unsurprising given that they are both local properties of rs-fMRI.

We note that other studies based on the correlation of brain imaging and genomics were able to use connectivity metrics as they utilized the Allen Human Brain Atlas (AHBA) which includes gene expression from >200 brain regions (PMID: 27574314, 31004051), allowing robust sample size to run connectivity analysis between regions in the corresponding fMRI/MRI dataset. However, there is no psychiatric disease gene expression with similar spatial resolution, making disease implications of such analyses limited. In fact, the ASD gene expression dataset used in this study is the largest one in the field to date.

There is not enough information provided regarding the resting state fMRI data analysis. Basic analytic decisions (eg. what were the head motion criteria/cutoffs) are not included.

We agree with the reviewer –and reviewer #1 for pointing out similar concerns - and apologize that the details were insufficient. We now include substantially more details of our methodology in the revised manuscript. We also note that we used Spearman's rank correlation between gene expression and fMRI measures. Spearman's rank correlation is likely to be robust to changes that will not dramatically alter fMRI values. Thus, as expected, we did not find a substantive alteration in the final set of genes in this revised manuscript when we included additional criteria that accounts for head motion. Please see our detailed responses to Reviewer #1 about analytic decisions as well as the updated Methods section of the revised manuscript.

Resting state fMRI and gene expression data were not available from the same subjects, limiting the interpretability of the presented results.

The reviewer is right that rs-fMRI and gene expression are from different subjects. To our knowledge, this is the case for nearly all studies in the brain activity – genomics field, except for the unique study recently published by our group (PMID: 33686299) that received attention by others (PMID: 33686296). But even in that study, we were only able to match brain activity and gene expression in one cortical region, which was made possible by a necessary brain resection in patients with epilepsy. We highlight these to emphasize both the practical limitations of the reviewer's request and the recognized scientific merit of studies that utilized both measures from different individuals.

Reviewer #3 (Remarks to the Author):

This manuscript is a detailed study integrating single cell gene expression data sets from neurotypical control and autistic patients (AUD) with regionally matched brain activity measurements obtained from fMRI datasets. The authors identify genes linked with brain activity that is disrupted in AUD patients. The gene sets are found to be enriched in voltage-gated ion channels and inhibitory neurons. An interesting result is that of the regions profiled primary visual cortex is seen to be the most affected region in AUD patients. The use of control and patient specific transcriptome profiles is unique in this study and is an important strength.

This work represents continued investigation in determining molecular correlates of functional imaging results, studies that have been of interest to the neuroimaging community for some time and with increased feasibility as brain wide deeper profiled molecular data sets have become available. The problem is important but challenging for several reasons including small effect sizes and highly correlated gene sets. This study has some of the same challenges but the approach and methods in the study are sound and well analyzed comparing disparate data sets. The authors have taken a reasonably comprehensive approach to setting up the problem and analysis.

We thank the reviewer for this positive feedback.

Some comments for consideration are:

1. Regions exhibiting significance differences in fALFF and ReHO are seen to be significant at a fairly weak level

($p < 0.05$). Can one be more specific about the distribution of these effects, perhaps even with exemplars compared across regions profiled?

We assume that the reviewer is referring to the weak effect size of fMRI signal differences between CTL-ASD. In our new analysis, while we found 4 ROIs different using fALFF, we only found BA20/37 to be significantly different between CTL-ASD in **both** fALFF and ReHo measurements (Supplementary Fig. 2a). We would like to clarify that since fMRI signal did not reliably differ between CTL-ASD, we identified gene-expression correlations with fMRI using only CTL fMRI samples (both in the previous and new analyses). We then identified genes correlated differentially between CTL_RNAseq – CTL_fMRI and ASD_RNAseq – CTL_fMRI.

We note that the resulting analysis yielded differentially correlated genes that were spatially altered. We highlight *PVALB* and *SCN1B* also in response to reviewer 1 (Fig 3g).

Fig. 3g: Gradient of CTL expressions (PC1), *PVALB*, and *SCN1B* gene expression. Barplot depicts correlation between *PVALB* and *SCN1B* gene expression with ReHo, fALFF, expression PC1 and expression PC2.

2. There are a very small number of genes found intersecting with previous studies such as Ricardi et al, and these are well known genes implicated with a variety of functions, e.g. *PVALB*, *SCN1B*, *SYT2*. How do the authors understand this limited intersection?

We thank the reviewer for this comment. As highlighted in the manuscript, the small overlap might be due to variation in cortical regions, type of fMRI measurements, and analyses. We would like to emphasize that the overlap is still highly significant ($p\text{-val} < 1e-10$) despite these variations (Fig. 3d, Supplementary Figure 5d).

3. In examining the relationship to development, it would be interesting to compare with the BrainSpan (www.brainspan.org) dataset which, although I believe is not single cell, contains a reasonably wide developmental trajectory and might provide insights connecting this work with previous studies.

We thank the reviewer for this suggestion and agree with their assessment. We now included the developmental analysis with BrainSpan. We included the novel analysis as supplementary figure 6c and we included the analysis information in the methods. This analysis showed similar trajectories for developmental gene clusters. For example, Adult cluster showed a similar pattern of gene expression increase also observed in the CTL samples but not in ASD samples (Fig 4a, Supplementary Figure 6c). We note that the Stable cluster behaves differently using the BrainSpan dataset. We reason that the Stable cluster is likely more heterogeneous than Adult or EarlyDev clusters and is probably composed of various genes that do not have a common expression pattern.

4. The differential correlation concept comparing control versus ASD is a strength of this study, and Fig 2. Illustrates the concept. It would be helpful to have some of the distribution properties of the associations found brought forward more transparently, perhaps a figure with ranking of genes by effect size.

We thank the reviewer for the comment. We now include a density plot as figure 3c to show the distribution of the DC genes' effect sizes. We amended the figure legend and the main text to reflect this change.

We amended the text as:

“For a $P < 0.01$, DC genes showed an effect size larger than 1.8, resulting in ~3% of the gene expressed in our data (Fig. 3c). Among the genes with highest effect size, we found *FILIP1*, a filamin A binding protein important for cortical neuron migration and dendrite morphology³⁸⁻⁴⁰, and *GABRQ*, a GABA receptor subunit highly expressed in von Economo neurons^{41,42}. In addition, the effect sizes of the DC genes calculated with fALFF and ReHo strongly correlate (Spearman Rank Correlation, $\rho = 0.54$, $p < 2.2 \times 10^{-16}$; Supplementary Fig. 5c) further confirming the reproducibility of the DC genes in two different rs-fMRI measurement.”

5. The ontological and protein association studies should be perhaps controlled with respect to background, at least for comparable brain function datasets. Generally network presentations such as those of Fig 3c are not particularly revealing unless substantiated with further evidence.

We thank the reviewer for pointing out this. We used brain expressed genes from Brainspan (N = 15585). We amended the method text to reflect this information. In the revised version, we decided to remove protein-protein interaction network because we agree that a) such results are challenging to interpret per Reviewer #1's comment and b) further evidence could be needed to substantiate such conclusions.

REVIEWER COMMENTS

Reviewer #2 (Remarks to the Author):

The authors have done a nice job conducting additional analyses in response to previous reviewer comments.

Reviewer #3 (Remarks to the Author):

The reviews and critiques of this paper were in depth and pointed out several areas for clarification and improvement. The authors have taken this revision very seriously and provided a substantial reworking of the material and new data analysis. I feel this addresses all or most of the concerns put forth and am comfortable with the analysis and present manuscript.

Reviewer #4 (Remarks to the Author):

Thanks for addressing previous comments. The following points remain unclear:

2a) Thanks for elaborating further on head motion, however authors do not provide any information on which metric they used for computing mean framewise displacement (for example, Power or Jenkinson etc?)

Also, usually head motion is included as an additional covariate at the second level.

2b) "Images are processed with a generalized linear model (GLM) to regress out: 1) global signal fluctuation, 2) physiological noise represented by white matter and CSF fluctuation, 3) fluctuation correlated with the 6 original affine head motion parameters (X/Y/Z/pitch/roll/yaw), 4) their first derivatives, squares, and squared derivatives, and 5) noise fluctuations captured from 5 components from aCompCor80."

I am a bit confused about all these steps – each one is a nuisance regression procedure and some of these are combined, but I have never seen all being used at once – usually studies report one such as for example compcor and then global signal regression as an alternative to test robustness of results. Especially using CompCor AND physiological noise from WM and CSF is redundant. Can authors please provide rationale for all these steps other than just having clicked all boxes in C-PAC?

Additional new comments:

- Please refrain from referring to autistic individuals as 'patients' in order to avoid ableist language (<https://www.liebertpub.com/doi/10.1089/aut.2020.0014>)

- I don't seem to find a table with demographic information for both samples – i.e., with average demographic information per sample. Also, how do the different sample compare to each other on the demographic information (given they were compared in subsequent analyses)

- The tense of language used in the methods preprocessing part switches between present and past tense.

- Given the analyses on developmental trajectories, authors should provide a supplementary figure describing the age distribution in the sample, as far as I remember ABIDE is densely sampled between 6-18 and has just very few data points above that age.

We would like to thank reviewer #4 for their additional comments on our manuscript. We are also pleased we were able to address all of the concerns of Reviewers #2 and #3.

Reviewer Comments:

Reviewer #2 (Remarks to the Author):

The authors have done a nice job conducting additional analyses in response to previous reviewer comments.

Thank you.

Reviewer #3 (Remarks to the Author):

The reviews and critiques of this paper were in depth and pointed out several areas for clarification and improvement. The authors have taken this revision very seriously and provided a substantial reworking of the material and new data analysis. I feel this addresses all or most of the concerns put forth and am comfortable with the analysis and present manuscript.

Thank you.

Reviewer #4 (Remarks to the Author):

Thanks for addressing previous comments. The following points remain unclear:

2a) Thanks for elaborating further on head motion, however authors do not provide any information on which metric they used for computing mean framewise displacement (for example, Power or Jenkinson etc?)

Jenkinson Framewise displacement was used. We have now updated the methods to include this information.

Also, usually head motion is included as an additional covariate at the second level.

Please see response to question 2b below.

2b) "Images are processed with a generalized linear model (GLM) to regress out: 1) global signal fluctuation, 2) physiological noise represented by white matter and CSF fluctuation, 3) fluctuation correlated with the 6 original affine head motion parameters (X/Y/Z/pitch/roll/yaw), 4) their first derivatives, squares, and squared derivatives, and 5) noise fluctuations captured from 5 components from aCompCor80."

I am a bit confused about all these steps – each one is a nuisance regression procedure and some of these are combined, but I have never seen all being used at once – usually studies report one such as for example compcor and then global signal regression as an alternative to test robustness of results. Especially using CompCor AND physiological noise from WM and CSF is redundant. Can authors please provide rationale for all these steps other than just having clicked all boxes in C-PAC?

We regress out all of these because we feel they are all sources of potential confounds when comparing ASD to CTL subjects, i.e. they may introduce artifactual correlation among ASD or CTL subjects, and thereby influence group differences that we may subsequently measure. To guard against that we aim to remove the potential confounds. For example, ASD subjects may have more difficulty keeping their head still throughout the scan compared to CTL subjects. Therefore, it is important to regress out head motion parameters and their variates, to suppress both linear and nonlinear artificial correlations. Also, we observed in our research [1, 2, 3] that no nuisance suppression approach is perfect, and we want to be quite confident that the differences between ASD and CTL subjects is real and not due to artifacts such as motion. Therefore, in certain cases we elected to use multiple estimates of a given nuisance, such as WM and CSF physiological nuisance, since the nuisance regressors computed through the techniques are not identical. While this may distribute the weight given to each regressor, we are not focused on nuisance regressor weight, but rather on thoroughly suppressing nuisance

artifacts from our signal (our primary goal). These are regressed out in succession beginning with HMP because we have found this produces the most reliable rsfMRI derived measures, which is important to us in this research.

Additional new comments:

- Please refrain from referring to autistic individuals as ‘patients’ in order to avoid ableist language (<https://www.liebertpub.com/doi/10.1089/aut.2020.0014>)

Thank you for that suggestion. We have removed “patients” from the text.

- I don’t seem to find a table with demographic information for both samples – i.e., with average demographic information per sample. Also, how do the different sample compare to each other on the demographic information (given they were compared in subsequent analyses)

We have now updated Table S1 to include all demographic information. Moreover, direct comparisons of additional demographics (not just age) are now included in the GitHub page.

- The tense of language used in the methods preprocessing part switches between present and past tense.

We have carefully checked this section and corrected it as needed.

- Given the analyses on developmental trajectories, authors should provide a supplementary figure describing the age distribution in the sample, as far as I remember ABIDE is densely sampled between 6-18 and has just very few data points above that age.

The age information of both the ABIDE and RNA-seq contributing individuals is now clearly noted in Table S1. Yes, there are many individuals in the range that the reviewer notes, but there are also older individuals. **We would also like to emphasize that we removed the variance explained by age (and other covariates) before we proceeded with analyzing the data to mitigate any effects of age (and other covariates) in both datasets.**

However, to directly address the reviewer’s concern, we also now show the direct comparison of ages in both sets of individuals for rsfMRI and RNA-seq. We now include these data in Supplementary Figure 8 as well as the GitHub page.

References:

1. Nguyen KP, Fatt CC, Treacher A, Mellema C, Cooper C, Jha MK, Kurian B, Fava M, McGrath PJ, Weissman M, Phillipps ML, Trivedi MH, Montillo A**. Patterns of Pre-Treatment Reward Task Brain

Activation Predict Individual Antidepressant Response: Key Results from the EMBARC Randomized Clinical Trial. *Biological Psychiatry*. 2021 [doi: 10.1016/j.biopsych.2021.09.011]

2. Raval, V, Nguyen, KP, Pinho, M, Dewey, RB, Trivedi, M, Montillo, AA. Pitfalls and Recommended Strategies and Metrics for Suppressing Motion Artifacts in Functional MRI. *Neuroinformatics*, 2022 [doi: 10.1007/s12021-022-09565-8]
3. Nguyen KP, Raval V, Treacher A, Mellema C, Yu FF, Pinho MC, Subramaniam RM, Dewey RB Jr, Montillo AA. Predicting Parkinson's disease trajectory using clinical and neuroimaging baseline measures. *Parkinsonism Relat Disord*. 2021 Apr;85:44-51. doi: 10.1016/j.parkreldis.2021.02.026.

REVIEWERS' COMMENTS

Reviewer #4 (Remarks to the Author):

Thanks for addressing my comments. Still I have to point out that these are alternative nuisance regression procedures and doing both aCompCor and mean regression of WM and CSF are redundant for example.

Also you did not answer whether head motion was included as a regressor in the second level analyses (which is different from preprocessing).

What I meant for the demographics is, it would be really helpful to get an idea of MEAN age, FIQ, sex distribution, mean of symptom measures (such as ADOS for example) etc to characterise the overall sample rather than each individual separately. Also, can you provide the information you put on GitHub? I cant seem to find it. I think it would also be helpful to incorpotae the sample differences/comparison in the main manuscript. Also, the second link in your manuscript doesn't work: https://github.com/DeepLearningForPrecisionHealthLab/AUTISM_rsfmri_ProcessingConnectivityExtractionAndSubjectMatching

We would like to thank reviewer #4 for their additional comments on our manuscript.

Reviewer #4 (Remarks to the Author):

Thanks for addressing my comments. Still I have to point out that these are alternative nuisance regression procedures and doing both aCompCor and mean regression of WM and CSF are redundant for example.

Thank you, we understand your point and do not think that redundant analyses are necessarily a bad thing, but rather bring additional assurance that the nuisance has been thoroughly suppressed.

Also you did not answer whether head motion was included as a regressor in the second level analyses (which is different from preprocessing).

Head motion was only included in preprocessing and fully addressed there. Since it was already addressed in preprocessing, we did not add it as a “regressed” covariate in secondary analysis. Several groups have attempted to leverage spatial heterogeneity of motion in the denoising process, e.g. through a second level covariate, yet these attempts have not been shown to consistently outperform other commonly used pipelines. This lack of improvement is most likely due to the fact that motion is highly correlated across voxels, and results in a drop in signal intensity across the entire brain parenchyma [Satterthwaite et al., 2013, Neuroimage, PMID: 22926292]. Meanwhile the pipeline we used included both global signal regression (GSR) and temporal censoring techniques which have been shown to be effective methods for minimizing the residual relationship between functional connectivity and motion artifacts. Combining GSR and censoring both minimizes the relationship between motion and connectivity, and simultaneously allows better detection of functional connectivity. [Satterthwaite, 2019, Human Brain Mapping, PMID: 29091315]

What I meant for the demographics is, it would be really helpful to get an idea of MEAN age, FIQ, sex distribution, mean of symptom measures (such as ADOS for example) etc to characterise the overall sample rather than each individual separately.

We apologize if this was not addressed sufficiently. Individual data were provided in the updated Table S1. We now also provide the mean of each demographic. These results are plotted in our GitHub page:

https://github.com/konopkalab/AUTISM_rsFMRI_GeneExpressionCorrelations/blob/master/rawdata/Demo_Pairs_Demographics.pdf

https://github.com/konopkalab/AUTISM_rsFMRI_GeneExpressionCorrelations/blob/master/rawdata/Demo_Pairs_Demographics_Abide.pdf

We also now provide each of these comparisons as updated Supplementary Figure 8.

Supplementary Figure 8. a) Pairwise comparison of demographic information containing biological and technical covariates for RNA-seq. In red: ASD subjects; in black: control. b) Pairwise comparison of demographic information containing biological and technical covariates for rs-fMRI. In red: ASD subjects; in black: control. c) Distribution of the age of the individuals who provided data for either RNA-seq or rs-fMRI studies.

Also, can you provide the information you put on GitHub? I cant seem to find it. I think it would also be helpful to incorporate the sample differences/comparison in the main manuscript.

Again, we apologize for this. Please see above GitHub links and supplemental figure 8 above.

Also, the second link in your manuscript doesn't work:

https://github.com/DeepLearningForPrecisionHealthLab/AUTISM_rsfMRI_ProcessingConnectivityExtractionAndSubjectMatching

We have now made this GitHub page public.